# Repression of varicella zoster virus gene expression during quiescent infection in the absence of detectable histone deposition

Jiayi Wang[1¤], Nadine Brückner[1], Simon Weissmann[2], Thomas Günther[2], Shuyong Zhu[1,3], Carolin Vogt[1,4], Guorong Sun[1], Rongrong Guo[1], Renzo Bruno[1], Birgit Ritter[1], Lars Steinbrück[1], Benedikt B. Kaufer[5], Daniel P. Depledge[1,3,4], Adam Grundhoff[2], Abel Viejo-Borbolla[1,3]*

1 Institute of Virology, Hannover Medical School, Hannover, Germany, 2 Leibniz Institute of Virology, Hamburg, Germany, 3 Excellence Cluster RESIST, Hannover Medical School, Hannover, Germany, 4 German Center for Infection Research (DZIF), Hannover, Germany, 5 Institute for Virology, Freie Universität Berlin, Berlin, Germany

¤ Current address: Qingdao Hospital, University of Health and Rehabilitation Sciences (Qingdao Municipal Hospital), Qingdao, China

* viejo-borbolla.abel@mh-hannover.de

**Data availability statement:** The RNAseq datasets generated and analysed in the current study are submitted to the European Nucleotide Archive (ENA) repository, with the following accession number: PRJEB68225. The ChIP-Seq and ATAC-Seq data for this study have been deposited in the ENA at EMBL-EBI under accession number PRJEB75685. All other

## Abstract

Varicella zoster virus (VZV) is a human-specific herpesvirus that establishes latency in peripheral neurons. The only transcripts detected in infected human trigeminal ganglia (TG) obtained shortly after death correspond to the VZV latency-associated transcript (VLT) and associated VLT-ORF63 splice variants. *In vitro* studies showed that VLT-ORF63 is translated into a protein (pVLT-ORF63) that induces VZV transcription. The mechanisms that lead to this restricted gene expression and the transition to lytic replication remain unknown, partly due to the difficulty of working with human neurons. In this study, we addressed whether the neuroblastoma-derived cell line SH-SY5Y could serve as a model to investigate the mechanisms that lead to repression of VZV gene expression followed by reactivation. VZV productively infected differentiated SH-SY5Y (dSH-SY5Y) whereas incubation with acyclovir (ACV) inhibited virus replication and induced a progressive repression of the virus. Upon removal of ACV there was production of viral particles in a subset of cells, while others contained non-replicating VZV genomes and VLT-containing transcripts for at least 20 days post-infection (dpi). Exogenous expression of VLT-ORF63 induced productive infection, suggesting that the non-replicating and repressed genomes remained functional. Interestingly, histone deposition was undetectable at VZV genomes in quiescently infected dSH-SY5Y cells, pointing to a potential novel mechanism leading to VZV repression in this neuronal setting.

## Author summary

Varicella zoster virus (VZV) establishes latent infection exclusively in human neurons. This phase of infection is characterized by restricted viral gene expression, lack of detectable protein and no infectious viral particles. Upon reactivation, infectious

data is within the manuscript or as supporting Information.

**Funding:** This work was funded by the Deutsche Forschungsgemeinschaft (DFG, German Research Foundation) under Germany's Excellence Strategy – EXC 2155 "RESIST"—project number 390874280 (A.V.B.) (https://www.resist-cluster.de/en/) and by the Deutsche Forschungsgemeinschaft (DFG, German Research Foundation) in the framework of the Research Unit FOR5200 DEEP-DV (443644894) through projects VI 762/4-1 (N.B., A.V.B.), GR 3318/5-1 (S.W., A.G.) and KA 3492/12-1 (B.B.K) (https://deep-dv.org/wp/). J.W., G.S., and R.G. were funded by a fellowship from the China Scholarship Council No. 201908370216, 201808230268 and 202303250019, respectively. J.W., N.B., S.Z., and R.G., were supported by the Hannover Biomedical Research School (HBRS) and the Center for Infection Biology (ZIB). The funders had no role in study design, data collection and analysis, decision to publish, or preparation of the manuscript.

**Competing interests:** The authors have declared that no competing interests exist.

viruses spread to new cells, causing disease. How VZV gene expression is repressed and regulated during latency is not well understood, partly due to the difficulty of obtaining large amounts of human neurons. Here, we established differentiated neuron-like SH-SY5Y cells as a model to study quiescent VZV infection in the presence of acyclovir, an inhibitor of VZV replication. Acyclovir incubation for six days led to very low VZV gene expression, no detectable viral protein and lack of infectious viral particles. Some of the cells contained viable VZV genomes for up to thirty days post-infection. We hypothesised that, similar to other herpesviruses, histones and their modifications would be present in VZV genomes during quiescent infection. However, we could not detect evidence of histone deposition in the majority of the VZV genomes, suggesting that they lacked chromatinization. These results point to other mechanisms restricting VZV gene expression while allowing genome persistence.

## Introduction

Varicella zoster virus (VZV) is a highly prevalent human pathogen that causes varicella during primary infection and herpes zoster upon symptomatic reactivation. A large percentage of elderly individuals also suffer post-herpetic neuralgia [1,2], the second most common type of neuropathic pain worldwide [3]. Moreover, VZV can also cause pneumonia, encephalitis, meningitis and vasculitis in some individuals [4,5].

VZV establishes latency in neurons of the peripheral nervous system and VZV DNA is detected in approximately 2–5% of sensory neurons in human trigeminal ganglia (TG), with an average of 5–7 copies of the viral genome per infected neuron [6–9]. Epidemiological data and clinical studies suggest that the virus can also establish latency and reactivate in autonomic neurons [10,11].

VZV latency is characterized by the persistence of the viral genome as an episome, restricted viral transcription, and the capacity of the virus to reactivate, leading to the production of new virions [12]. The VZV latency transcript (VLT) is the only consistently detected VZV transcript in human TGs obtained at short post-mortem intervals. Several TGs also contain VLT splice variants that incorporate the open reading frame 63 (ORF63) sequence (VLT-ORF63) [13,14]. VLT is encoded antisense to ORF61, the VZV homolog of herpes simplex virus (HSV) infected cell protein 0 [13]. *In vitro* studies with VZV-latently infected human induced pluripotent stem cell (iPSC)-derived neurons (termed HSN) confirmed the expression of VLT during latency, while VLT-ORF63 transcripts were only detected following the incubation of the cells with reactivation stimuli [14]. Translation of VLT-ORF63-1 results in a protein (pVLT-ORF63) that induces widespread VZV gene expression *in vitro* [14]. These results suggest that VLT-ORF63 plays a role during reactivation rather than latency.

The cellular and viral processes leading to VZV latency and reactivation are still unclear. In particular, the kinetics and mechanisms of VZV genome repression are not known. Foetal human neurons have been employed to study VZV neuropathogenesis *ex vivo* or xenotransplanted in severe combined immunodeficiency (SCID) mice [4,15,16]. Access to these human cells is scarce, and not permitted in certain countries, complicating their use as a model in many laboratories. As an alternative, human neurons can be derived from embryonic and adult stem cells as well as from iPSCs to study VZV latency and reactivation [17]. Human stem cell-derived neurons treated with acyclovir (ACV) one day prior to infection and infected with low multiplicity of infection (MOI) in the presence of ACV during 6 days support quiescent VZV infection [18]. A similar model had been established for HSV by Wilcox and Johnson and is widely employed to study HSV latency and reactivation, with variations in the exposure time to ACV [19–25].

In an alternative model, infection of stem cell-derived human neurons through the axonal end resulted in a phenotype reminiscent of latency [18]. Interestingly, both models showed a similar phenotype, characterized by very low genome-wide viral gene expression, no detectable protein translation and no viral particle production. The axonal model of infection was also employed to determine other aspects of VZV latent infection, including the reactivation potential of the vaccine Oka strain and the expression profile of the VZV latency transcript (VLT) *in vitro* [14,26].

The derivation of human neurons from stem cells is expensive and time consuming. In addition, it is difficult to obtain sufficient neuronal cells for mechanistic experiments and to study the kinetics of VZV gene repression during establishment of latency. Furthermore, the obtained neuronal cultures tend to be heterogenous [27–29] and the starting precursor culture and the differentiation method employed determine the percentages and types of derived neuronal cells [30]. There is thus an unmet need for an expandable neuron-like model that allows the study of VZV repression and reactivation.

To address this, we here examined the utility of the SH-SY5Y cells for the study of VZV repression prior to the establishment of latency. SH-SY5Y is a subclone of a neuroblastoma cell line obtained from a bone marrow biopsy [31]. SH-SY5Y cells can be differentiated into mature neuron-like cells by different protocols and are commonly employed to study neurological processes and diseases [32–34]. They have also been employed to study the neurotropism of several viruses including VZV [35,36], HSV and HSV-derived vectors [37,38], as well as human cytomegalovirus [39]. While laboratory adapted and clinical VZV strains productively infect differentiated SH-SY5Y cells (dSH-SY5Y) [35,36], latent infection and reactivation has not been studied with these cells.

Here we established a model to study VZV repression and de-repression employing dSH-SY5Y cells. Our results suggest that a progressive repression of VZV gene expression occurs upon ACV incubation. Non-replicating viral genomes and transcripts from the VLT locus were detected in a small percentage of dSH-SY5Y cells up to 20 days post-infection (dpi). Ectopic expression of pVLT-ORF63 induced productive VZV infection. Interestingly, the bulk of VZV genomes in non-productively infected cells was not occupied by histone H3 and lacked detectable nucleosomes. Despite the apparent absence of repressive chromatin, however, we found only a subfraction of genomes to be in an accessible state, a finding which was in accord with the observed low levels of transcripts throughout the viral genome and the absence of detectable viral protein and virus production. These results, together with the expandable nature and robust differentiation of SH-SY5Y provide an opportunity to study the mechanisms leading to VZV repression and de-repression in human neuron-like cells.

## Results

### Differentiation of SH-SY5Y cells into neuron-like cells

Non-differentiated SH-SY5Y cells contain a mixture of neuronal and epithelial precursor cells. To obtain differentiated SH-SY5Y (dSH-SY5Y) cells with characteristics of human neurons, we modified a successful differentiation method [40]. A schematic representation of the protocol is shown in Fig 1A. The main modification from the original protocol was the detachment of neuron-like cells with collagenase followed by seeding onto Matrigel-coated plates, while the epithelial-like cells remained attached onto the original well.

After 18 days of differentiation, the dSH-SY5Y neuron-like cells had a smaller cell body than the original cells and long, branched neurite projections connecting with the surrounding cells (Fig 1B). Furthermore, from 18 days post differentiation (dpd), the dSH-SY5Y cells expressed several proteins found in mature neurons, including microtubule-associated protein

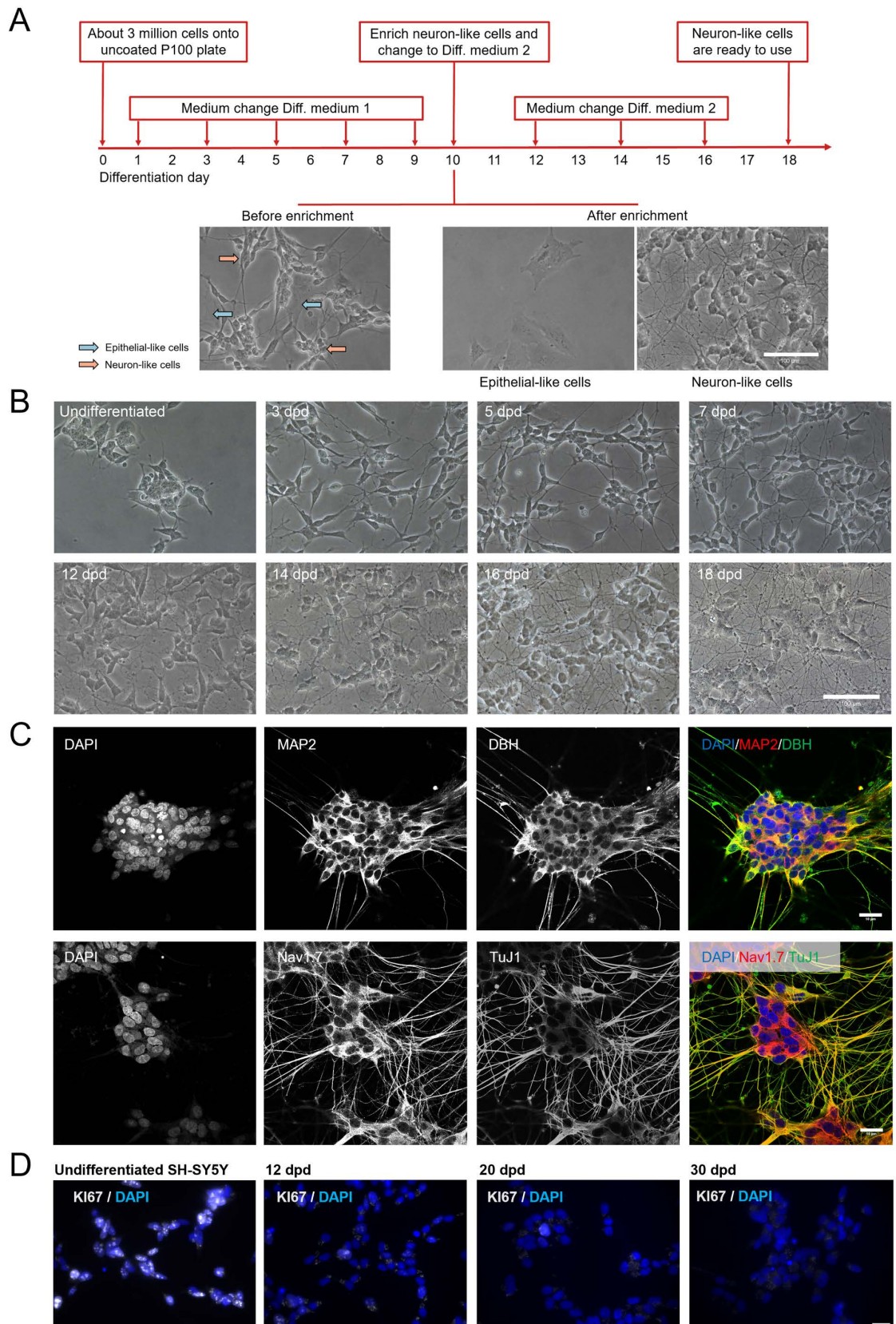

**Fig 1. Successful differentiation of SH-SY5Y into neuron-like cells.** (A) Schematic representation of the differentiation process showing representative pictures of SH-SY5Y before treatment with collagenase (left) at day 10 post-differentiation, epithelial-like

cells that remained in the well (middle) and neuron-like cells after seeding onto a new Matrigel-coated well. Scale bar: 100 μm. (B) Representative pictures of SH-SY5Y during the differentiation process. Scale bar: 100 μm. (C) Representative pictures showing expression of neuronal markers in dSH-SY5Y at 18 dpd. Scale bar: 10 μm. (D) Representative images showing KI67 expression in SH-SY5Y during the differentiation process. Scale bar: 20 μm. The results shown in this figure are representative from three independent differentiations. Abbreviations: dpd, days post-differentiation; MAP2, microtubule-associated protein 2; DBH, dopamine beta-hydroxylase; Nav1.7, voltage-gated sodium channel Nav1.7; TuJ1, beta-III-tubulin antibody.

2 (MAP2), β-III-tubulin (Tuj1), Nav 1.7, and dopamine beta hydroxylase (DBH) (Fig 1C). We did not detect DAPI positive cells lacking neuronal markers, suggesting that the majority of cells had a neuron-like phenotype. We then tested whether dSH-SY5Y cells underwent active mitosis by performing KI67 staining. In undifferentiated SH-SY5Y cells, 56% of the cells were KI67 positive. This reduced to 14% at 12 dpd, 3% at 20 dpd, and 1% at 30 dpd (Fig 1D), indicating that dSH-SY5Y cells were predominantly post-mitotic at 20 dpd.

These results indicate that the modified protocol resulted in successful neuronal differentiation of SH-SY5Y cells.

## VZV reporter virus replicates efficiently in neuron-like SH-SY5Y cells

To follow the infection and spread of VZV in dSH-SY5Y cells, we employed a recombinant bacterial artificial chromosome (BAC) VZV pOka strain [41] expressing RFP fused to immediate early ORF63 and GFP fused to the leaky-late gene ORF11 (termed v63R/11G, Fig 2A). The expression of the fluorescent proteins allowed us to observe the progression of VZV productive infection.

We infected dSH-SY5Y cells with cell-free v63R/11G at an MOI of 0.001, and detected RFP and GFP positive cells at 3, 5 and 12 dpi. The number of RFP and GFP positive cells increased over time (Fig 2B). In some cells we observed RFP expression in the absence of detectable GFP. This could be due to lack of synchronous infection throughout the culture or because VZV gene expression did not proceed further in those cells. Similarly, the viral genome copy number increased from an average of 2 copies per cell at 1 dpi to an average of 50 copies per cell at 6 dpi (Fig 2C). The transcripts of immediate early genes ORF4 (IE4), ORF61 (IE61) and early gene ORF68 (glycoprotein E, gE) [42] also increased over time (Fig 2D-F). These results showed that v63R/11G efficiently replicates in dSH-SY5Y, in line with previous observations [35,36].

## The duration of ACV incubation determines the level of repression of VZV gene expression in dSH-SY5Y cells

A previous study showed that pretreatment of human stem cell-derived neurons with ACV for 24 hours followed by low MOI VZV infection in the presence of ACV for 6 days leads to a phenotype reminiscent of latency [18]. Therefore, we employed the same procedure in an attempt to establish VZV quiescent infection and reactivation in dSH-SY5Y cells. We also employed different ACV incubation times to determine whether this would impact VZV repression. We pretreated dSH-SY5Y cells for 24 hours with ACV and then infected them at an MOI of 0.001 in the presence of the drug during 2, 3, 4, 5 or 6 dpi (Fig 3A). We monitored the cells twice a day for 30 dpi to detect ORF63-RFP and ORF11-GFP expression, indicative of productive VZV replication. We observed a negative correlation between the number of days the infected-cells were incubated with ACV and the time post-ACV removal when ORF63-RFP- and ORF11-GFP-positive cells were detected (Fig 3B and 3C). Since we detected some cells positive for ORF63-RFP with undetectable ORF11-GFP signal, we decided to represent in the graph wells that had at least one cell positive for ORF63-RFP, irrespective of GFP signal. At

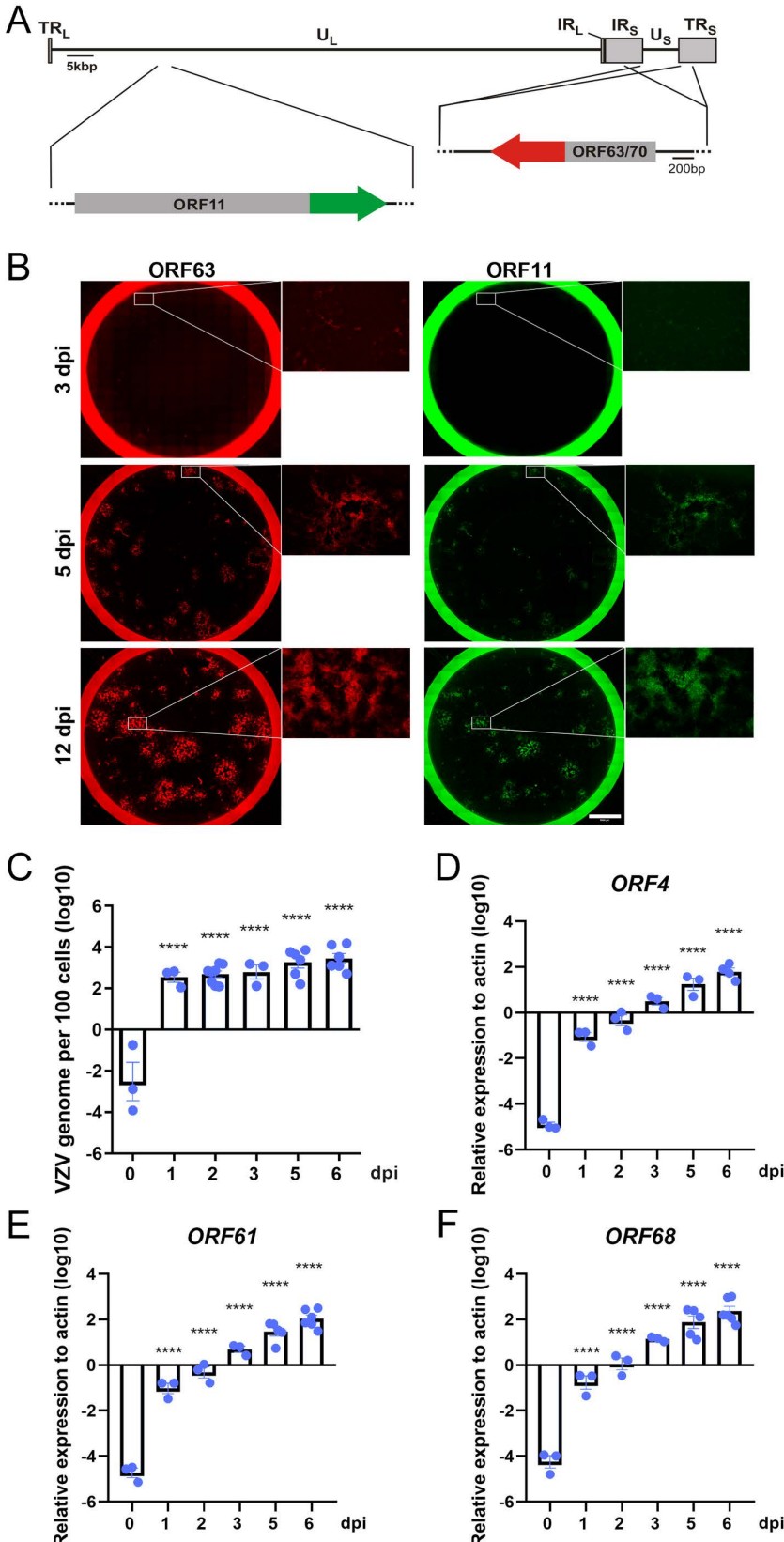

**Fig 2. VZV productively infects dSH-SY5Y cells.** (A) Schematic representation of the recombinant v63R/11G showing the terminal repeats long and short (TRL and TRS, respectively) the unique long and short regions (UL and

US, respectively) and the internal repeat long and short (IRL and IRS, respectively). ORF63 and ORF70 correspond to duplications of the same ORF, with ORF63 located in the IRS and ORF70 in the TRS. (B) Images showing spread of v63R/11G in dSH-SY5Y cells. (C-F) Graphs showing VZV genome copies n=3-9 (C) and relative expression of VZV genes, n=3-6 (D-F) at different times post-infection. The results in (C-F) are from 3 biological replicates. Values are presented as mean ± s.e.m. Statistical comparisons between infected cells at 0 dpi and rest of the conditions were performed using one-way ANOVA on log-transformed raw data to stabilize variances and improve normality. P > 0.05 (ns), P ≤ 0.05 (*), P ≤ 0.01 (**), P ≤ 0.001 (***), P ≤ 0.0001 (****). Abbreviations: dpi, days post-infection.

30 dpi, only 1 well (1.4% of wells) had ORF63-RFP-positive cells when cells had been incubated with ACV for 6 dpi. However, this increased to 45.7% of wells in cells incubated with ACV for 5 dpi.

## Incubation with ACV for 5 days results in a repressive phenotype that can be spontaneously reversed

We next focused on dSH-SY5Y cells incubated with ACV for 5 dpi (Fig 4A). The samples containing cells expressing ORF63-RFP and ORF11-GFP have replicating VZV and were termed "R", while those lacking the fluorophores were termed non-replicating, "NR" (Fig 4B). The expression of ORF63-RFP and ORF11-GFP also correlated with the presence of VZV gE (Fig 4C).

We quantified the expression of *ORF61* and *ORF68* during acute infection as well as at several dpi following incubation with ACV in samples lacking detectable ORF63-RFP (Fig 4D and 4E). We also analyzed cells expressing ORF63-RFP at different days post-release of repression (dpr) by ACV (Fig 4D and 4E). The expression of both *ORF61* and *ORF68* was higher during acute infection than in the other conditions and it increased with time, indicating productive viral replication and virus spread. By contrast, the expression of both genes was very low in the presence of ACV and following ACV removal. The viral gene expression in wells that contained ORF63-RFP/ORF11-GFP positive cells, indicative of VZV replication, was 10–20 times higher than in those lacking ORF63-RFP/ORF11-GFP positive cells but lower than in acute infection, and also increased with time (Fig 4D and 4E).

We also quantified VZV genome copy numbers as a surrogate of VZV replication (Fig 4F). The number of viral genomes increased in the cells infected without ACV, while in those exposed to ACV for 5 days, the level of viral genomes decreased with time. To determine whether cells lacking detectable ORF63-RFP and ORF11-GFP expression contained the viral genome, we performed *in situ* hybridization (DNAscope) in wells lacking detectable expression of these fluorophores and detected VZV DNA (VLT locus) at 12 and 20 dpi. The virus genome was detected in about 5% of the cells that were incubated with ACV for 5 days (Fig 4G and S1 and S2 Tables). The low number of viral genomes detected by *in situ* hybridization combined with qPCR results, suggest that there was no ongoing viral replication in these cells.

Finally, seeding of dSH-SY5Y cells expressing ORF63-RFP and ORF11-GFP on top of ARPE19 cells led to productive infection of these epithelial cells (Fig 4H), demonstrating the presence of infectious viral particles. These results suggest that incubation with ACV for 5 days leads to two phenotypes, one characterized by cells containing viral genomes that produce infectious virus and another one by cells that maintain non-replicating viral genomes.

## Less than 5% of dSH-SY5Y cells incubated with ACV during 6 days maintain non-replicating viral genomes for at least 30 dpi

The presence of ACV 1 day prior to VZV infection and during 6 dpi led to a near complete repression of VZV, with 98.6% of VZV infected wells containing cells lacking ORF63-RFP and ORF11-GFP expression for up to 30 days (Figs 3C, 5A and 5B). Similarly, we could not detect

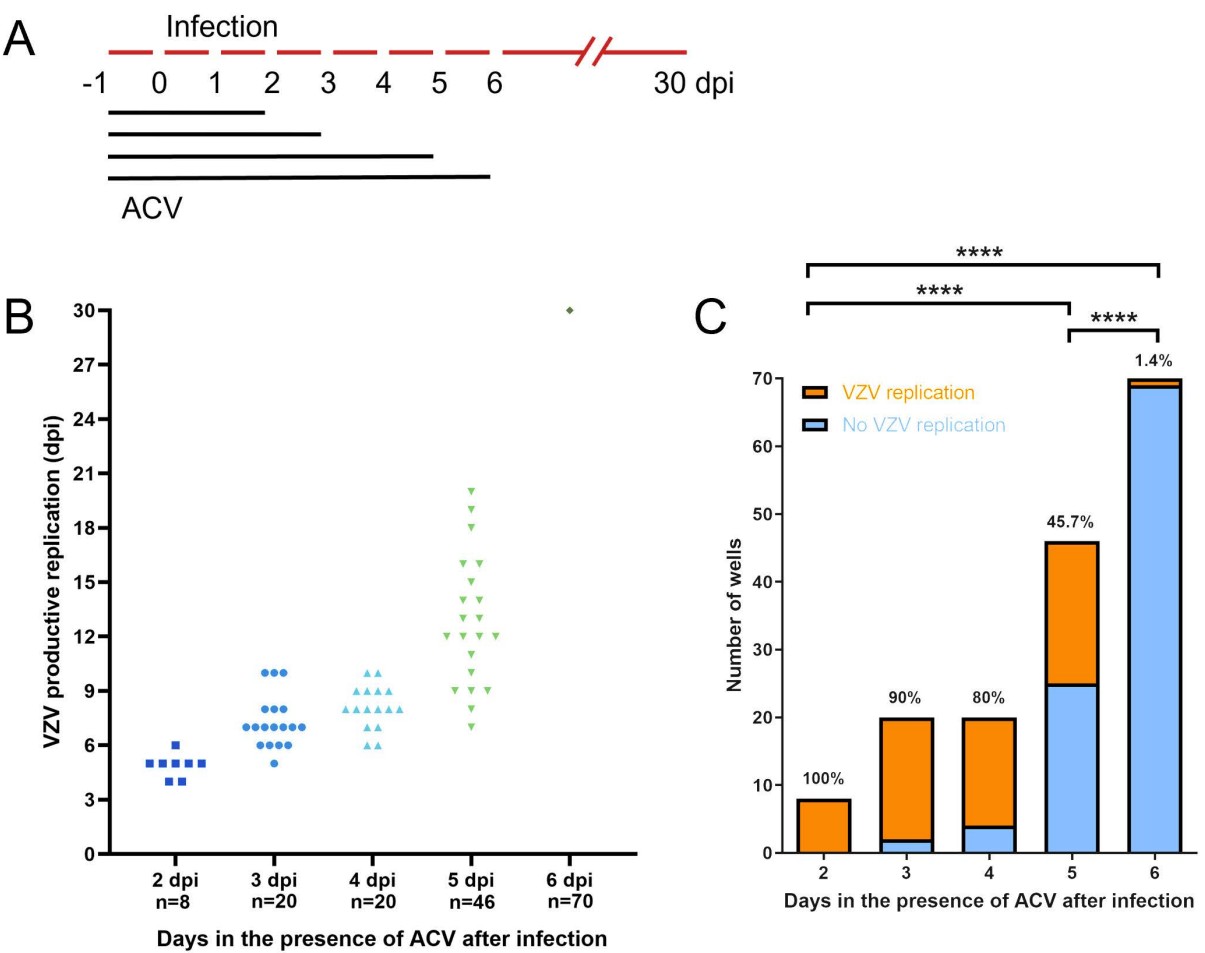

**Fig 3. Incubation with ACV progressively represses VZV in dSH-SY5Y cells.** (A) Schematic representation of the experiment. (B) Graph showing the day post-infection when wells containing ORF63-RFP/ORF11-GFP positive dSH-SY5Y cells were detected following incubation with ACV for 2-6 days. Each symbol represents one well containing ORF63-RFP/ORF11-GFP positive cells. "n" refers to the number of wells infected with VZV for each condition. (C) Graph showing the number of wells containing ORF63-RFP/ORF11-GFP positive (orange) and negative (blue) dSH-SY5Y cells following incubation with ACV for 2-6 days. The percentage on top of each column indicates the percentage of wells with ORF63-RFP/ORF11-GFP positive dSH-SY5Y cells at the end of the experiment (30 dpi). The data corresponds to 4, 8 and 14 biological experiments (differentiation plus infection) for samples treated with ACV for 2-4 dpi, 5 dpi and 6 dpi, respectively. The "n" in (B) indicates the number of analyzed wells per condition. Statistical comparisons were performed using one-way ANOVA on log-transformed raw data to stabilize variances and improve normality. P > 0.05 (ns), P ≤ 0.05 (*), P ≤ 0.01 (**), P ≤ 0.001 (***), P ≤ 0.0001 (****). Abbreviations: ACV, acyclovir; dpi, days post-infection.

viral proteins gE and IE4 by WB at different dpi following 6 days incubation with ACV (Fig 5C). The expression of VZV *ORF4*, *ORF61* and *ORF68* was much lower in ACV incubated cells than in acute infected cells at 6 dpi and decreased further over time following ACV removal (Fig 5D–F). We detected viral DNA by qPCR in the inoculated wells at 6, 12, 16, 20 and 30 dpi, with DNA copy numbers consistently averaging 1 genome copy or less per cell (Fig 5G).

We detected VZV DNA (VLT locus) at 12 and 20 dpi in about 4.4% of cells that were incubated with ACV for 6 days (Fig 5H and S3 and S4 Tables). These results suggested that a low number of dSH-SY5Y cells infected with VZV in the presence of ACV for 6 dpi maintain VZV genomes with very low gene expression, lack of detectable protein and virus production, potentially reflecting a quiescent state.

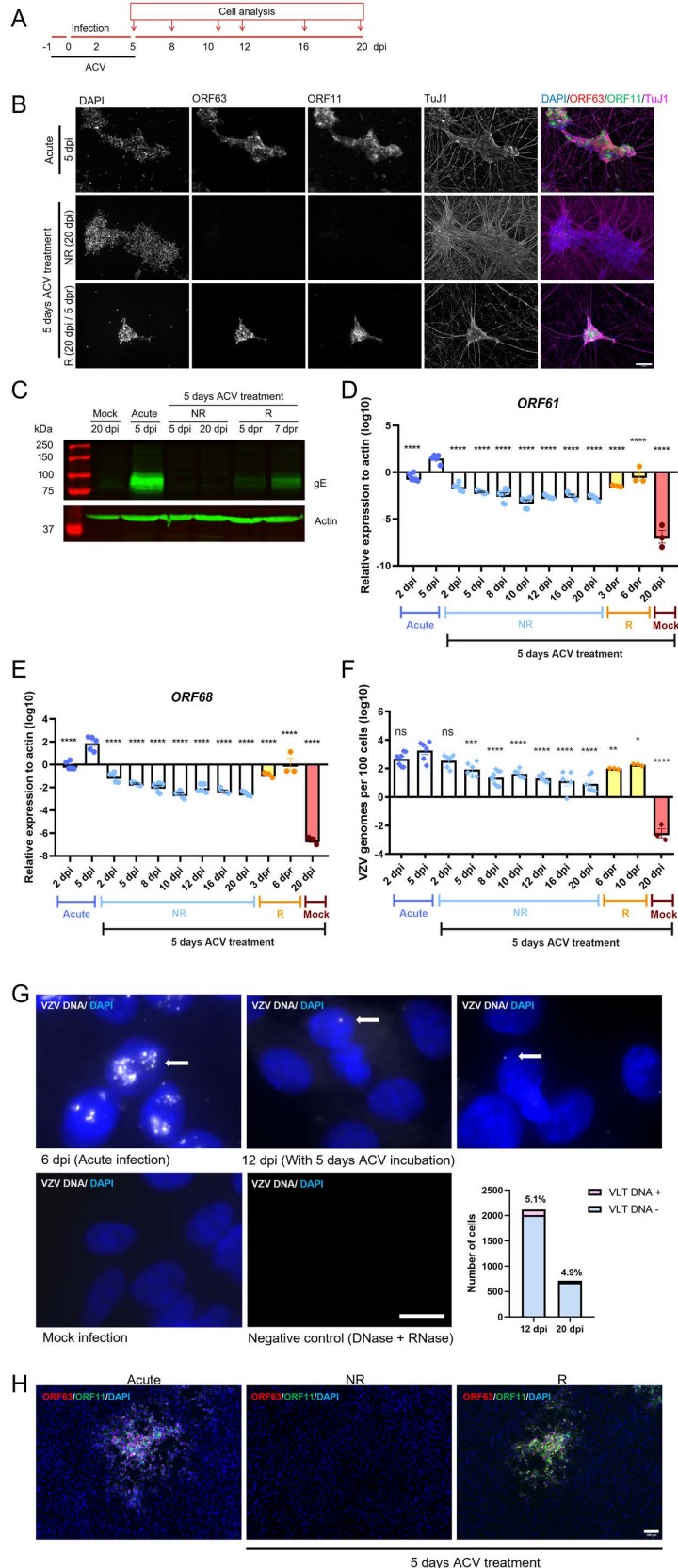

**Fig 4. Incubation with ACV during 5 days results in a repressive phenotype that can be released.** (A) Schematic representation of the experiment. The arrows indicate days when cells were collected for further analysis. (B)

Representative images showing dSH-SY5Y cells infected with v63R/11G in the absence (acute) or presence of ACV for 5 dpi and lacking or containing ORF63-RFP/ORF11-GFP positive dSH-SY5Y cells. The dSH-SY5Y cells were labelled with an anti-TuJ1 antibody and the nuclei were stained with DAPI. Scale bar: 100 μm. (C) Western blot detecting VZV gE (top blot) and actin (bottom blot) in dSH-SY5Y cell lysates obtained from mock- or v63R/11G-infected cells without ACV (acute) or with 5 days incubation with ACV. (D-F) Relative gene expression of VZV genes, n=3-9 (D,E) and quantification of VZV genomes, n=3-9 (F) in mock- or v63R/11G-infected dSH-SY5Y cells in the absence (acute) or presence of ACV for 5 dpi. Statistical comparisons between acute infection at 5 dpi and other infection conditions were performed using one-way ANOVA on log-transformed raw data to stabilize variances and improve normality. P > 0.05 (ns), P ≤ 0.05 (*), P ≤ 0.01 (**), P ≤ 0.001 (***), P ≤ 0.0001 (****). (G) Detection of VZV genomes by *in situ* hybridization in dSH-SY5Y infected with v63R/11G in the absence (acute) or presence of ACV for 5 days in wells where no ORF63-RFP/ORF11-GFP positive cells were detected. DAPI was used to stain nuclei. The white arrows point to the VZV genomes. The data corresponds to 41 and 14 fields of view acquired randomly in for 12 and 20 dpi, respectively, obtained from three biological experiments. Scale bar: 10 μm. (H) Detection of ORF63-RFP/ORF11-GFP in ARPE19 cells incubated with dSH-SY5Y cells (infected in the presence of ACV for 5 days) from wells lacking (NR) or containing (R) ORF63-RFP/ORF11-GFP-positive cells. DAPI was used to stain nuclei. In all panels NR refers to "non-replicating" VZV, while R refers to "replicating" VZV, determined by the expression of ORF63-RFP/ORF11-GFP. Abbreviations: ACV, acyclovir; dpi, days post-infection. The results in (D-F) are from 3 biological replicates. The images in (G) are representative from more than six samples. Values are presented as mean ± s.e.m.

We repeated these experiments with a BAC-derived pOka strain VZV expressing GFP instead of ORF57 (pOka-Δ57-GFP, S1 Fig). This virus was previously generated by Paul (Kip) Kinchington (Accession number PP378487; [43]). We obtained similar repression of VZV in the presence of ACV for 6 days, suggesting that the obtained results were not strain specific.

## Ectopic VLT-ORF63 expression induces VZV replication and virus production in infected dSH-SY5Y cells incubated with ACV for 6 days

We next examined whether addition of drugs previously used as reactivation stimuli could induce de-repression of VZV after 6 days incubation with ACV. We tested LY294002, an inhibitor of phosphoinositide 3-kinase (PI3K) and suberanilohydroxamic acid (SAHA), an inhibitor of histone deacetylases, at 8 dpi (2 days after removal of ACV) (Fig 6A). LY294002 has been previously shown to induce VZV reactivation [18], while SAHA induces reactivation of Kaposi's sarcoma-associated herpesvirus (KSHV) [44]. Incubation with LY294002 or SAHA slightly increased VZV gene expression without detectable ORF63-RFP and ORF11-GFP protein, lack of infectious virus, and led to cell death after 4 days of incubation (Fig 6B and 6C). We repeated these experiments at 34°C since VZV spreads more efficiently at this temperature after reactivation induced with LY294002 [18], and monitored the cells daily from the day of LY294002 or SAHA addition until ten days post-addition. We did not observe any cells expressing ORF63-RFP or ORF11-GFP after incubation with LY294002 or SAHA at 34°C.

Ectopic VLT-ORF63 expression induced transcription of VZV IE, E, and L genes in latently VZV-infected HSN, suggesting that the pVLT-ORF63 fusion protein is involved in the transition from latency to lytic infection [14]. Therefore, we addressed whether the ectopic expression of VLT-ORF63 could induce VZV reactivation and production of infectious virus in the dSH-SY5Y cells infected in the presence of ACV for 6 days. We incubated quiescently infected dSH-SY5Y cells with SAHA and LY294002, or transduced them with lentiviruses expressing VLT-ORF63 or GFP (S2 Fig). The VLT-ORF63 lentivirus induced VZV protein expression and virus spread, monitored by ORF63-RFP and ORF11-GFP positive cells in about half (21/41) of the wells (Fig 6D and 6E), while the eGFP control lentivirus or the treatment with SAHA and LY did not (Fig 6B–D).

A single nitrocellulose membrane was used to detect protein expression sequentially (S3 Fig, blots on left side). VLT-ORF63 or ORF63-RFP proteins were detected in acutely infected cells, in "NR" (probably expressed from the VLT-ORF63 lentivirus) and "R" samples. ORF63,

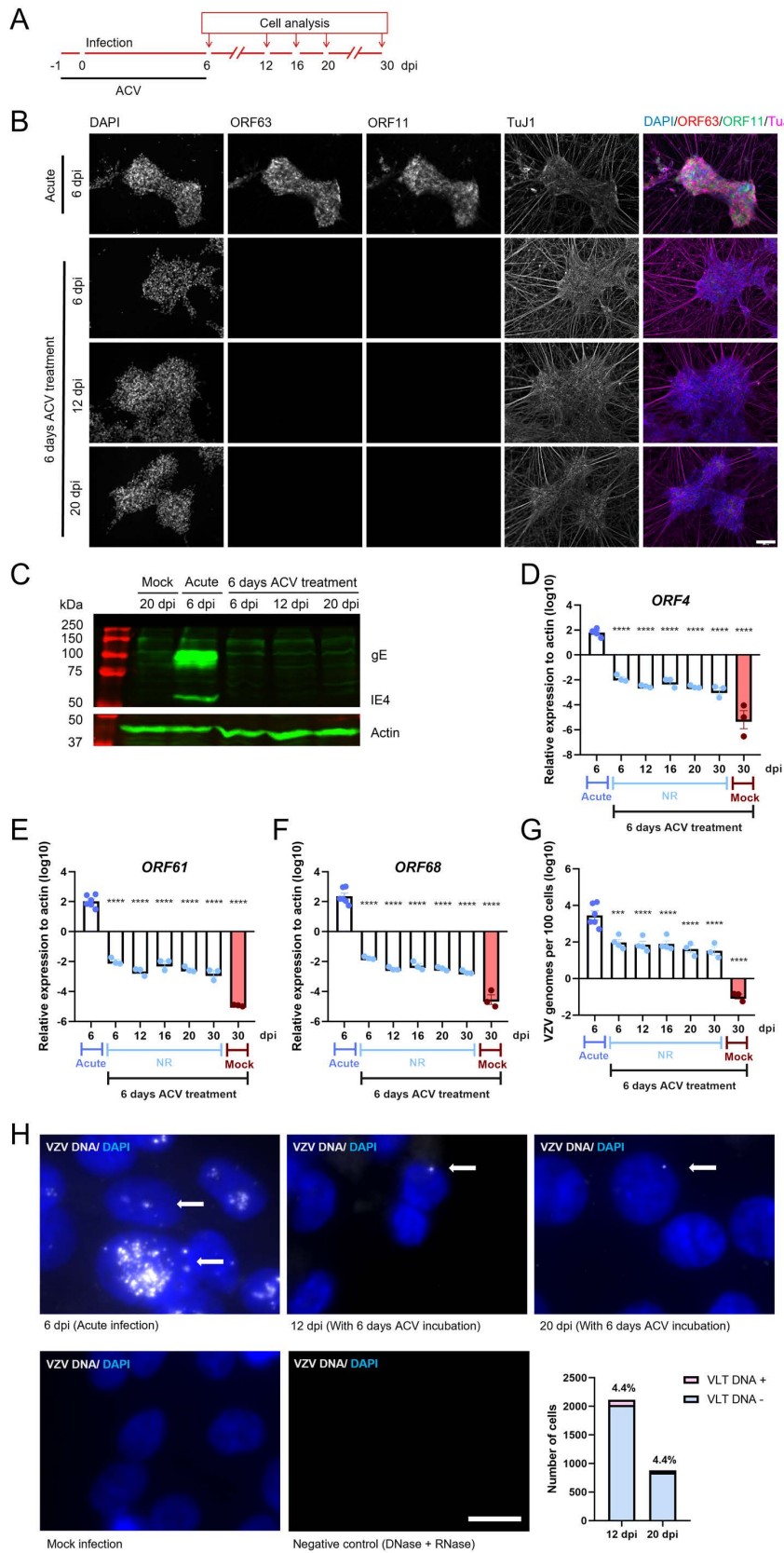

**Fig 5. A small percentage of dSH-SY5Y cells incubated with ACV for six days maintain non-replicating viral genomes for up to 20 days.** (A) Schematic representation of the experiment. The arrows indicate days when cells

were collected for further analysis. (B) Representative images showing dSH-SY5Y cells infected with v63R/11G in the absence (acute) or presence of ACV for 6 dpi and lacking or containing ORF63-RFP/ORF11-GFP positive dSH-SY5Y cells. The dSH-SY5Y cells were labelled with an anti-TuJ1 antibody and the nuclei were stained with DAPI. Scale bar: 100 μm. (C) Western blot detecting VZV gE and IE4 (top blot) and actin (bottom blot) in dSH-SY5Y cell lysates obtained from mock- or v63R/11G-infected cells without ACV (acute) or with 6 days incubation with ACV. (D-G) Relative gene expression of VZV genes, n=3-6 (D-F) and quantification of VZV genomes, n=3-6 (G) in mock- or v63R/11G-infected dSH-SY5Y cells in the absence (acute) or presence of ACV for 6 dpi. Statistical comparisons between acute infection at 6 dpi and other infection conditions were performed using one-way ANOVA on log-transformed raw data to stabilize variances and improve normality. P > 0.05 (ns), P ≤ 0.05 (*), P ≤ 0.01 (**), P ≤ 0.001 (***), P ≤ 0.0001 (****). (H) Detection of VZV genomes (grey dots) by *in situ* hybridization in dSH-SY5Y cells infected with v63R/11G in the absence (acute) and presence of ACV for 6 days in wells where no ORF63-RFP/ORF11-GFP positive cells were detected. DAPI was used to stain nuclei. The white arrows point to the VZV genome. The data was obtained from 41 and 16 random images for 12 and 20 dpi, respectively, from three biological experiments. Scale bar: 10 μm. The graph on the right shows the percentage of cells containing the VZV genome following infection in the presence of ACV during 6 days and subjected to *in situ* hybridization at 12 and 20 dpi Abbreviations: dpi, days post-infection. The results in (D-G) are from 3 biological replicates. Values are presented as mean ± s.e.m.

gE and ORF11-GFP were detected only in acutely infected cells and "R" samples, while eGFP was also observed in cells transduced with eGFP control lentivirus. The lower intensity of the VLT-ORF63/ ORF63-RFP signal observed in the "NR" sample compared to the "R" one could indicate that lack of de-repression was due to lower level of pVLT-ORF63 expression. Considering the similar size of VLT-ORF63 and ORF63-RFP proteins, another nitrocellulose membrane loaded with the same samples was incubated with an anti-RFP antibody to confirm the expression of ORF63-RFP in acutely infected cells and "R" samples (S3 Fig, blots on right side).

These results show that pVLT-ORF63 releases the repression on the VZV genome in dSH-SY5Y cells incubated with ACV for 6 days, highlighting that the infection was not abortive.

## Incubation with ACV leads to low level genome-wide VZV transcription in infected dSH-SY5Y cells

We next analyzed the VZV transcriptome in dSH-SY5Y cells infected in the presence of ACV during 3, 4, 5 and 6 dpi at different times post-ACV removal. We detected transcripts across the VZV genome, although of low magnitude (Fig 7A and 7B). The expression level negatively correlated with ACV incubation time. Whether any of these transcripts corresponds to mature RNA that could be translated is unknown at present. We detected transcription across the VLT exons in acutely infected cells and at 2 days post-ACV removal following 5 and 6 days incubation with ACV (S4 Fig). VZV genome-wide expression was also reported when infecting human stem cell-derived neurons in the presence of ACV or through the axonal end [18,26]. We also performed RNA *in situ* hybridization (RNAscope) with a probe that binds VLT and VLT-ORF63 transcripts [14] and detected these transcripts in the cytoplasm of about 1% of dSH-SY5Y cells at 12 and 20 dpi (following incubation with ACV for 6 days) (Fig 7C). This corresponds to about 20–25% of cells containing the viral genome (S5 and S6 Tables). These results suggest that a small number of dSH-SY5Y cells maintained the viral genome without active replication and expressing VLT or VLT-ORF63.

## Only a minority of VZV genomes in quiescently infected dSH-SY5Y cells are accessible while the bulk of genomes lack detectable histone deposition

We performed ChIP-seq and ATAC-seq analyses of ACV-treated dSH-SY5Y cells 13 days after infection to elucidate the chromatin status of resident VZV genomes. These experiments were performed with v63R/11G and the parental BAC-derived pOka strain lacking any

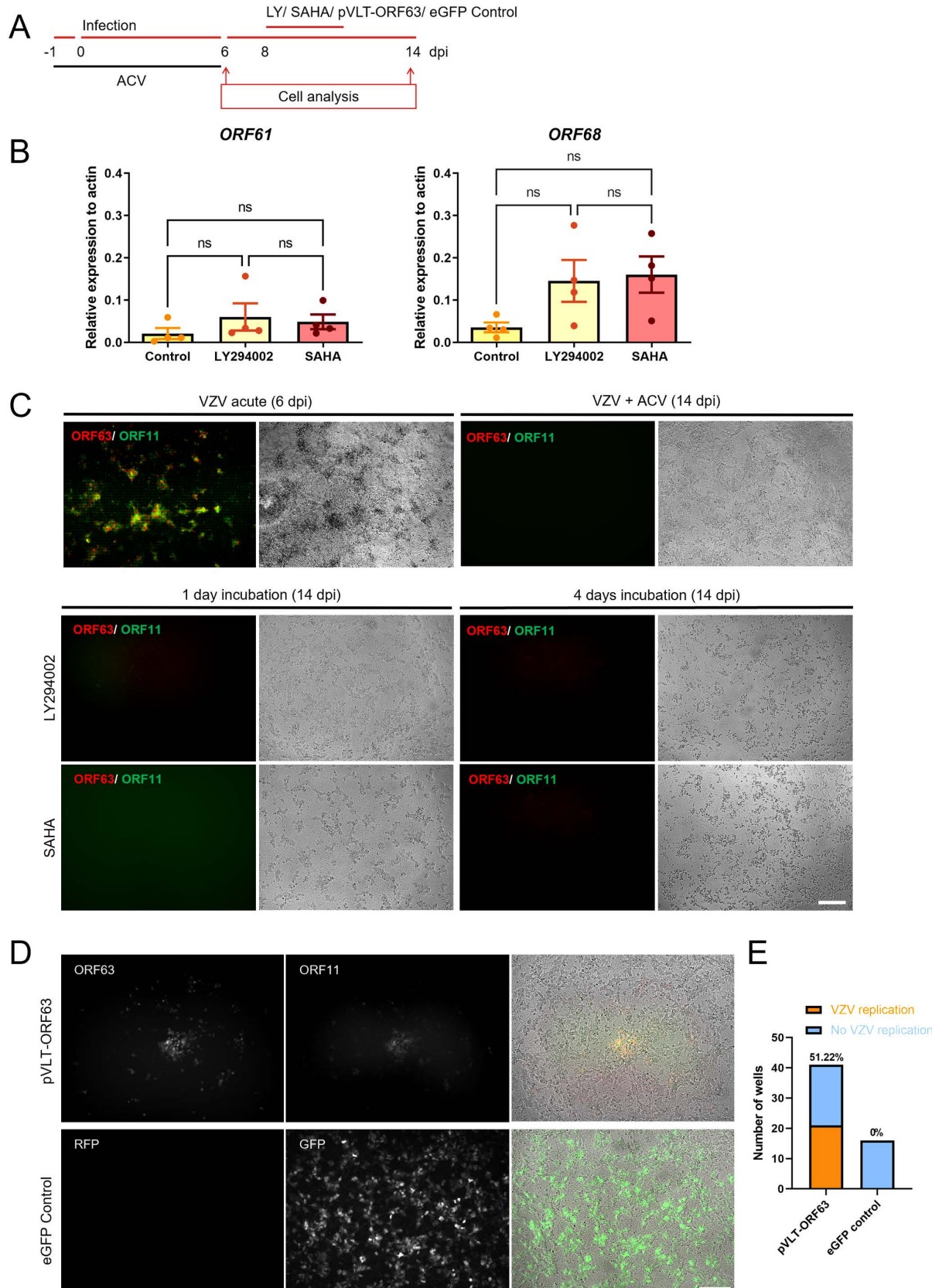

**Fig 6. Exogenous expression of pVLT-ORF63 results in production of infectious VZV from cells exposed to ACV during 6 days.** (A) Schematic representation of the experiment. The arrows indicate days when cells were analyzed. (B) Relative VZV gene expression in dSH-SY5Y cells

infected with v63R/11G in the presence of ACV for 6 days and incubated with LY294002 or SAHA at 8 dpi, n=3-4. The results are from 3 biological replicates. Values are presented as mean ± s.e.m. (C) Representative images of dSH-SY5Y infected cells in the absence (acute) or presence of ACV during 6 days and incubated or nor with LY294002 or SAHA at 8 dpi during 4 days. The pictures were taken at 6 dpi for acute infected cells and at 14 dpi for the samples treated with ACV. (D) Representative images of dSH-SY5Y cells infected with v63R/11G in the presence of ACV for 6 days and transduced with a lentivirus expressing VLT-ORF63 (top) or eGFP (bottom). The left and middle panels show direct fluorescent expression of ORF63-RFP, ORF11-GFP, RFP and GFP, while the right panels show the merge of the fluorescent channels with phase-contrast images. Scale bar: 100 μm. (E) Graph showing the percentage and number of wells containing ORF63-RFP/ORF11-GFP positive dSH-SY5Y cells (representing repli-cating virus), following transduction with lentivirus expressing VLT-ORF63 or eGFP. Abbreviations: ACV, acyclovir; dpi, days post-infection.

fluorophores, termed WT [41]. Mapping to the human genome confirmed the fidelity and functionality of reagents and experimental protocols. As examples, the top and center panels in Fig 8A show coverage tracks of two regions on chromosome 17 and 19 that encompass loci enriched for facultative and constitutive heterochromatin marks (H3K27me3 and H3K9me3, respectively), but also contain euchromatic promoter regions decorated by activation-associated H3K4me3 and H3K27ac marks. S5A Fig shows the average read densities for ChIP-seq and ATAC-seq samples across all annotated human transcriptional start sites (TSSs). As expected, H3K27ac and H3K4me3 signals were strongly enriched in the +/−2.5 kb flanking regions, with locally decreased coverage indicative of a nucleosome-free region at the TSS in the center. Conversely, ATAC-seq densities exhibited marked peaks flanking the position of the +1 nucleosome.

Unexpectedly, although VZV was highly covered by input reads, we did not observe any significant histone modification patterns across the viral genome (lower panel in Fig 8A). Pan-H3 ChIP-seq coverage was also markedly low compared to the host genome, suggest-ing that most VZV genomes were not chromatinized. In support of this conclusion, Fig 8B shows a relative enrichment analysis of the viral genome compared to positive and negative host regions for each of the analyzed histone marks. For this purpose, we determined aver-age ChIP-seq signals by calculating enrichment in the most significantly called host peak regions (positive control regions, left panel) relative to a set of randomly selected negative host regions (baseline value = 1; center panel) and compared these to input-normalized values from windows shifted across the viral genome (right panel). Importantly, this approach accounts for genome size differences, sequencing depth, and viral copy numbers by normal-izing read counts to window size and input reads. As anticipated, the magnitude of positive control ChIP-signals varied between individual antibodies, with the greatest and lowest values observed for the H3K4me3 and pan-H3 antibodies, respectively (note that low signals are to be expected for pan-H3, as overall nucleosome density does not exhibit high variability across host regions). As shown in the right panel of Fig 8B, ChIP-seq signals in the viral genome were consistently below one (i.e., the value assigned to the negative control regions), likely reflecting the fact that histone-free genomes do not elicit either specific or unspecific signals and contribute to input only.

In contrast to ChIP-seq, ATAC-seq produced appreciable coverage across the entire viral genome (bottom panel and track in Fig 8A). Nevertheless, quantitative analyses of input-normalized ATAC-seq signals demonstrates that viral ATAC-seq signal levels, though signifi-cantly above background (p=1.2$^{-48}$, one-sided heteroscedastic t-test), represented only 6% of those seen in positive host regions (S5B Fig). This observation suggests that, while most VZV genomes are not histone-decorated, only a minority are accessible to Tn5 transposase during ATAC-seq. Since we obtained similar results with parental pOka and v63R/11G, our results indicate that the obtained results are not strain specific.

To explore whether ATAC-seq-accessible VZV genomes are nucleosome-bound, we ana-lyzed the insert size distribution of paired read fragments. Human-mapping reads exhibited

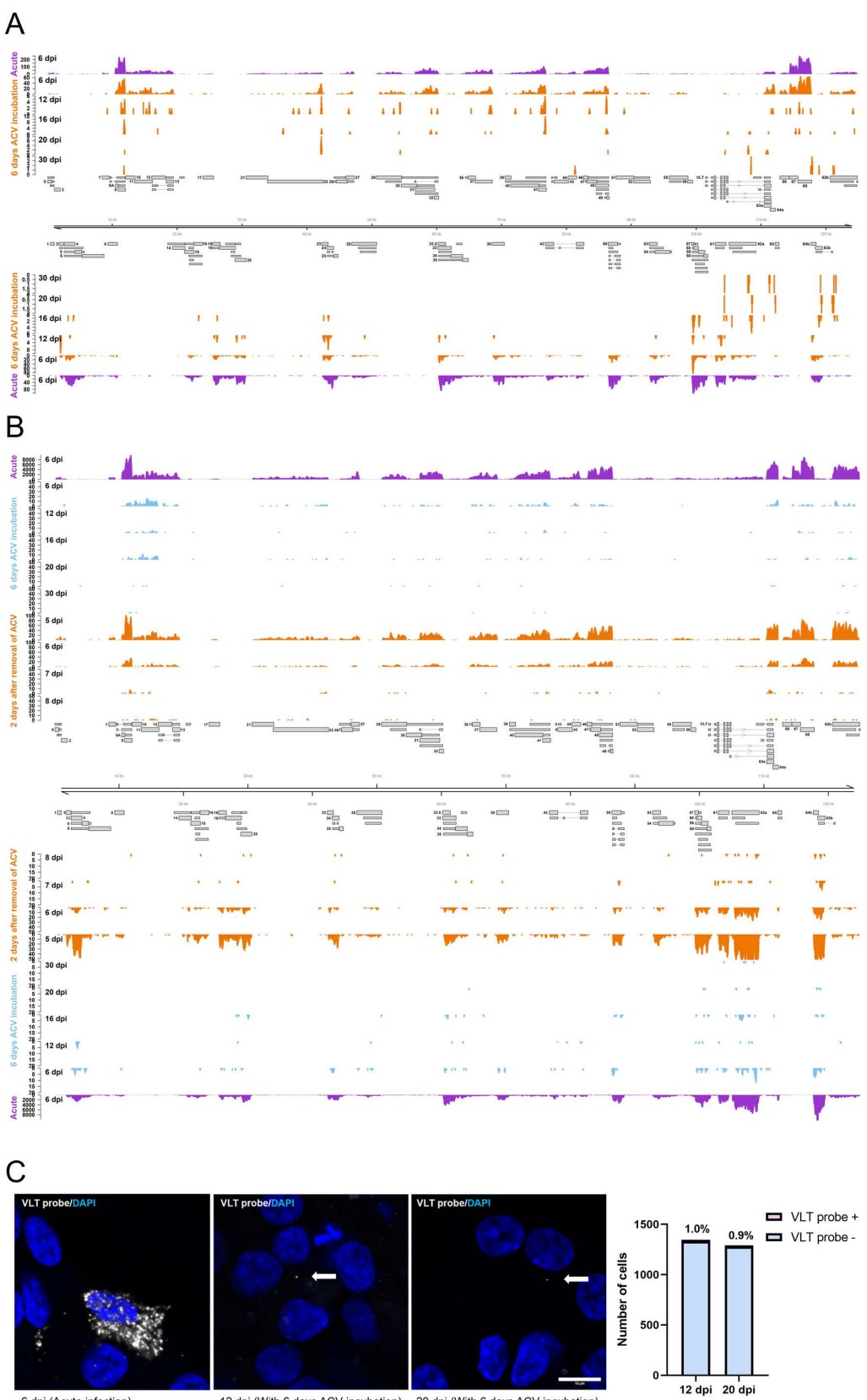

**Fig 7. Incubation with ACV leads to low level genome-wide VZV transcription in dSH-SY5Y cells. (A,B)** Genome-wide transcription profiles of dSH-SY5Y cells infected with v63R/11G in the absence (acute, violet) and presence of

ACV for 6 dpi (orange, A; blue, B) or for 3, 4, 5 and 6 dpi (orange, B). Bulk RNA-Seq was performed at different times post-infection, as labelled. Transcription from both DNA strands is shown with the depth of coverage labelled on the y-axis. A representation of the VZV genome and all encoded transcripts is shown. The samples labelled as 12, 16, 20, 30 dpi correspond to transcriptomic analyses of dSH-SY5Y cells infected with VZV in the presence of ACV during the first 6 days of infection. Total RNA was extracted on days 12, 16, 20 and 30 post-infection and subjected to RNA-sequencing. The samples labelled as 5, 6, 7 and 8 dpi correspond to cells that were infected in the presence of ACV during 3, 4, 5 and 6 days and were processed for RNA extraction two days after removal of ACV. (C) Detection by *in situ* hybridization of VZV mRNA (grey dots) with a probe that detects VLT and VLT-ORF63 transcripts in dSH-SY5Y infected in the absence (acute) or presence of ACV for 6 days in wells where no ORF63-RFP/ORF11-GFP positive cells were detected. DAPI was used to stain nuclei. The white arrows point to the transcripts. Scale bar: 10 μm. The graph on the right shows the percentage of cells expressing VLT or VLT-ORF63 transcripts following infection in the presence of ACV during 6 days and subjected to *in situ* hybridization at 12 and 20 dpi. Abbreviations: ACV, acyclovir; dpi, days post-infection.

distinct peaks corresponding to nucleosome free regions (NFRs) and mono-, di-, tri-, and oligonucleosomes, consistent with periodic DNA wrapping (~160 bp per nucleosome; Fig 8C, top panel). This pattern persisted when read numbers were subsampled to match VZV read counts (Fig 8C, middle panel). However, VZV-mapping reads showed only a single clear NFR peak, suggesting that ATAC-seq-accessible viral genomes are largely devoid of nucleosomes.

This surprising lack of histones on VZV genomes prompted us to investigate alternative explanations. Possible factors include insufficient viral genomes to confidently assess chromatinization via ChIP-seq, the induction of latency using ACV treatment, or the use of non-replicating dSH-SY5Y cells, which may poorly support chromatinization following herpesvirus infection. To rule out these possibilities, we treated dSH-SY5Y cells with ACV for 24 hours, infected them with low MOI of a BAC-derived KSHV (KSHV BAC16, [45]) in the presence of ACV, and performed ChIP-seq 13 days post-infection. KSHV was chosen as a model due to its well-characterized chromatinization and histone modification profiles. Although KSHV has lower sensitivity to ACV than to other antivirals [46], the dosage used in this study (100 μM) is still expected to inhibit its lytic replication. Furthermore, low-dose KSHV infection allows ChIP-seq analyses of viral copy numbers similar to or lower than VZV.

Using the same quantification strategy for histone modifications, we observed strong enrichment of H3K27me3 and H2AK119Ub on KSHV genomes (S5C Fig). H3K4me3 and H3K27ac showed a broader spread of ChIP signal at KSHV genomic windows, reflecting the expected highly focused enrichment of these marks at active viral promoters. KSHV ChIP-seq signals of H3K9me3 and pan-H3 were close to quantified signals of these marks at human negative regions, aligning with previous reports of low H3K9me3 levels at KSHV genomes post-infection [47] and the even distribution of pan-H3 across chromatinized DNA. Coverage tracks comparing host and KSHV regions highlight the enrichment of active marks (H3K4me3, H3K27ac) at latency-associated promoters (e.g., ORF75, ORF73, miRNA cluster and the global distribution of polycomb repressive complex associated histone marks H3K27me3 and H2AK119Ub in an anticorrelate pattern to H3K4me3 (S5D Fig). Despite lower viral copy numbers relative to VZV (S5E Fig), KSHV genomes showed detectable chromatin patterns, indicating that ChIP-seq is sensitive enough to reveal chromatinization even at low viral copy numbers, provided the genomes are chromatinized.

## Discussion

How neuronal cells repress and regulate VZV gene expression prior to and during latency and how this repression is released upon reactivation is not known. This is partly due to the difficulty of performing experiments such as ChIP-seq that require large number of cells with human neuronal models that support latency and reactivation. Here, we attempted to

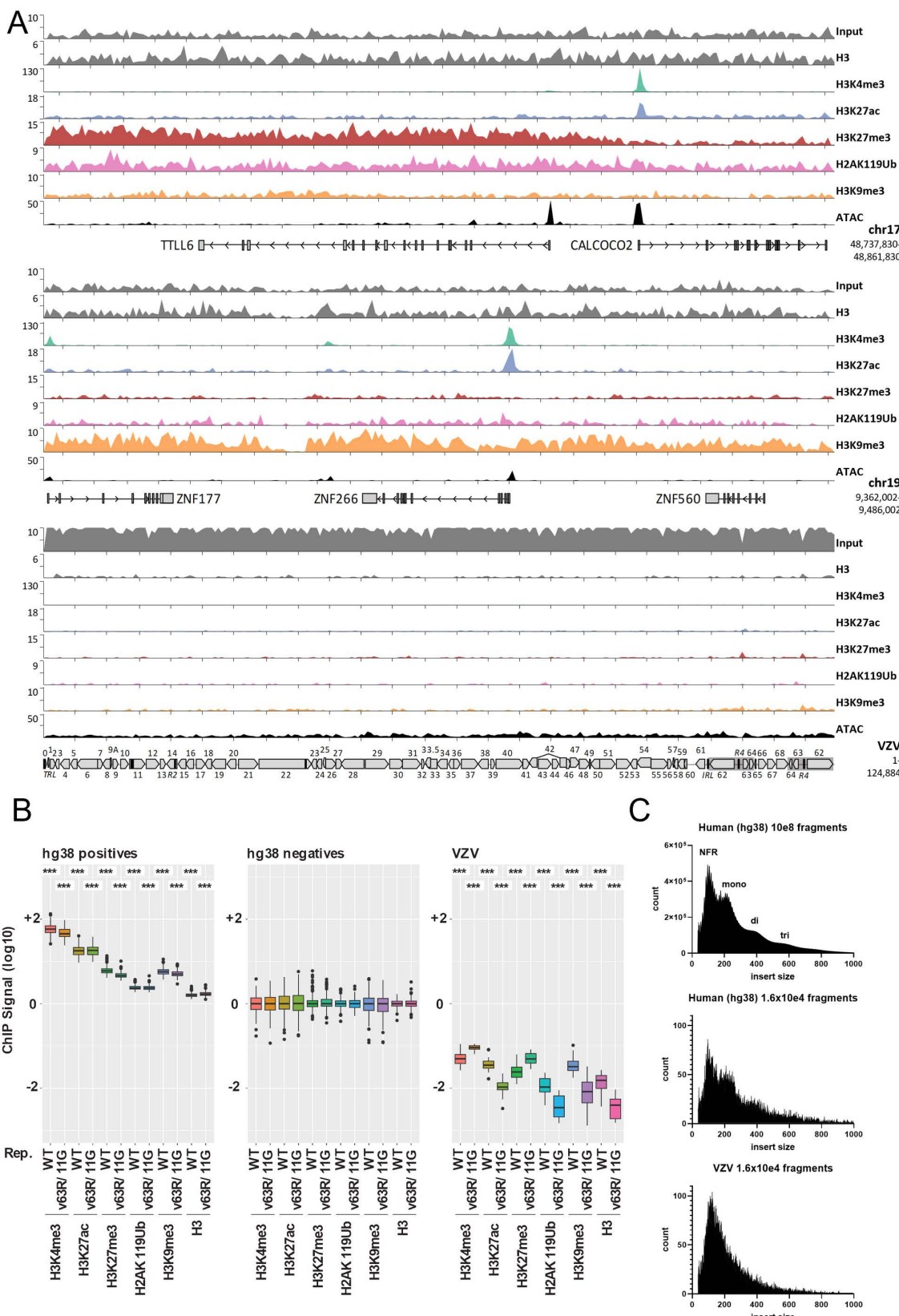

**Fig 8. ChIP-seq and ATAC-seq analysis of differentiated SH-SY5Y cells quiescently infected with VZV at 13 days post-infection.**
(A) Read density coverage tracks of histone marks and ATAC-seq signal on two host loci and VZV, determined by ChIP-seq. Coverage

track of VZV shows data from dSH-SY5Y cells latently infected with VZV pOKA WT as a representative of the two independent virus strains used in ChIP experiments. (B) Input-normalized quantification of ChIP-seq signals in a 10 kb sliding window across the VZV genome (VZV, right panel), relative to the 200 most significantly enriched host regions (hg38 positives, left panel) and an equal number of size matched, randomly selected host control loci (hg38 negatives, center). Signals observed in host control regions were set to 1 (10E0). Experiments were independently performed with BAC-derived, wild-type VZV (WT; VZV pOKA WT) and a BAC-derived VZV carrying fluorescent markers (v63R/11G; VZV pOKA Bac ORF11-GFP/ORF63-RFP). ($P > 0.05$ (n.s), $P \leq 0.05$ (*), $P \leq 0.01$ (**), $P \leq 0.001$ (***) Wilcoxon-Mann-Whitney-Test of indicated sample compared to corresponding hg38 negatives). (C) Fragment size distribution of ATAC-seq data. Size distribution of human reads (hg38) are shown for all aligned reads (top panel) and randomly subsampled to match numbers of aligned reads to VZV (middle panel). Lower panel shows fragment size disruption of reads aligned to VZV (VZV pOKA WT). Peaks indicative of nucleosome free regions (NFR), mononucleosomes (mono), dinucleosomes (di) and trinucleosomes (tri) are indicated within the upper graph.

establish a human neuronal model with dSH-SY5Y cells to investigate how VZV is repressed and maintained in a quiescent state.

dSH-SY5Y cells are commonly employed in neurobiological research [48–50], in neuroinfection [35,36,51,52] and support full VZV replication and cell-to-cell spread [35,36,53], but a quiescent state has not been previously reported.

Differentiation of SH-SY5Y cells was successful as shown by the expression of neuronal markers, as well as by lack of cell division. Two different recombinant VZV replicated and spread in dSH-SY5Y, in line with previous reports [35,36,54]. We employed ACV to establish a non-productive infection as previously done with stem cell-derived neurons [18]. Longer ACV incubation times drastically reduced the frequency of productive replication after ACV removal. These results, together with the progressive reduction in gene expression across the whole VZV genome upon removal of ACV, suggest that following VZV entry in dSH-SY5Y cells, there was a progressive repression of viral gene expression that correlated with the duration of ACV treatment. One interesting observation was the different phenotype when cells were treated with ACV for 5 days in comparison with 6 days. Incubation with ACV for 5 dpi repressed VZV but allowed spontaneous de-repression in about half of the wells following removal of the drug. In contrast, repression after 6 days of ACV treatment was nearly complete, with only 1.4% of wells containing productively infected cells at 30 dpi.

Approximately 4.4% of individual neuron-like cells treated with ACV for 6 dpi retained the viral genome for at least 20 dpi. The fact that very low genome copies were normally detected in these cells suggests that there was no DNA replication and the infection could be abortive. However, about 1% of these repressed cells – corresponding to approximately 20–25% of cells harboring the genome – expressed transcripts across VLT exons at 12 and 20 dpi. Since the RNAscope probe employed detects both VLT and VLT-ORF63 transcripts, we cannot conclude which one is expressed in these neuron-like cells. However, these results suggest the establishment of restricted gene expression in a reduced number of dSH-SY5Y cells and indicate that the infection in the presence of ACV during 6 days was not abortive, at least in these cells. This was supported by the production of infectious virus upon exogenous expression of VLT-ORF63.

We also observed a reduction in the level of viral gene expression over time when analyzing the VZV transcriptome. The transcriptome profile of VZV in dSH-SY5Y cells incubated with ACV was similar to that of acutely infected cells, although with much lower expression. Previous results employing stem cell-derived neurons also found that following ACV incubation, or upon axonal infection, the transcription profile of VZV did not mirror the transcriptome obtained in human TG following decades of latent infection [18,26]. In these reports the expression of VLT was not investigated since this transcript had not been discovered yet [13]. In another report, axonal infection of human iPSC-derived neurons with cell-free VZV pOka

strain led to expression of VLT and non-detectable expression of ORF63 by RT-qPCR, suggesting that a latent phenotype was achieved, although the genome-wide transcription profile of VZV was not analyzed [14].

As has been shown for HSV [55,56], regulation of VZV latency is probably mediated by a combination of immune and epigenetic mechanisms. We hypothesized that repressive histone modifications were responsible for the phenotypes observed after long-term ACV incubation in dSH-SY5Y cells. Gary and colleagues showed the presence of the euchromatin mark H3K9ac on VZV genomes obtained from postmortem human TG ganglia [57], indicating that H3 is present on the VZV genome in this setting. Surprisingly, however, we could not detect any significant enrichment of histones (H3 and modifications as well as H2AK119Ub) on the viral genome of infected dSH-SY5Y cells treated with ACV during 13 days. These results were obtained with two different recombinant VZV strains, the parental pOka and v63R/11G, indicating that they were not due to the artificial modification of the viral genome. Lack of signal was not due to low sensitivity of the ChIP-seq assays, since we observed chromatinization of KSHV genomes despite these being less abundant than those of VZV. The KSHV results also suggest that the lack of VZV chromatinization is not due to the cellular model employed nor the use of ACV. We do not understand how VZV genomes persist in the absence of chromatinization.

Our observations strongly suggest that the bulk of VZV genomes in ACV-treated cells are nucleosome-free and consequently cannot be silenced by transcriptional repressors recruited via histone marks. This conclusion is supported by the lack of nucleosomal pattern observed following analysis of the ATAC-seq results. Nevertheless, considering that only a small proportion of viral genomes produced ATAC-seq signals, we suspect that repression may, at least in part, be the consequence of reduced accessibility of viral genomes to transcription factors and/or the transcriptional machinery. While the underlying mechanisms will doubtlessly require further investigation, we consider sequestration or entrapment of genomes in subnuclear compartments such as, for example, phase-separated promyelocytic leukemia nuclear bodies (PML-NBs) or stalled replication compartments as potentially contributing factors. PML-NBs surround HSV-1 genomes during latency and contribute to their chromatinization by imposing histone H3.3 marks on the viral genome [58,59]. We did not detect histone H3 on VZV genomes. Moreover, PML-NB impairs HSV-1 reactivation [60]. This latter phenomenon was observed only upon induction of PML expression by type I IFN. Whether such a restriction occurs in our model is currently unknown.

There is also the possibility that at least some of the persisting VZV genomes may be partially or fully protected by capsid proteins, e.g., in virions trapped at the nuclear envelope or in PML bodies [61,62]. Likewise, it is possible that histone-independent recruitment of repressors such as IFI16 or SMC5/6 could contribute to transcriptional repression, as shown for other viruses including HSV-1 [63–68]. Another confounding factor could be DNA methylation, although this epigenetic mark does not seem to be relevant during HSV-1 latency [69,70]. Apart from the presumably inactive genomes, we also do not know to what extent the minority of VZV genomes that are accessible in ATAC-seq assays contribute to the observed phenotypes. Since these genomes appear to be globally accessible, it is tempting to speculate that they may represent the source of the observed low-level genome-wide transcription patterns. However, at present we also cannot exclude the possibility that transcription and reactivation originates from a very small subfraction of chromatinized genomes that is below the detection limit of our ChIP-seq assays.

Finally, we presently do not know to what extend our findings apply to *in vivo* latency in sensory neurons or other models of *in vitro* infection. Therefore, more research is warranted to understand how VZV is repressed upon infection of neuronal cells.

## Materials and methods

### Cells

Neuroblastoma-derived SH-SY5Y (ATCC-CRL-2266) and epithelial ARPE19 (ATCC-CRL-2302) cells were maintained in a humidified incubator at 37 °C with 5% $CO_2$. Undifferentiated SH-SY5Y cells were cultured in DMEM/Nutrient mixture F-12 Ham medium (Sigma) with 15% Fetal bovine serum (FBS, Sigma), supplemented with penicillin–streptomycin (Cytogen) and L-glutamine (Cytogen). ARPE19 cells were cultured in DMEM/Nutrient mixture F-12 Ham medium with 8% FBS, supplemented with penicillin–streptomycin and L-glutamine.

### Viruses

We previously fused the monomeric red fluorescent protein (mRFP) to the C-terminus of ORF63/70 in the pOka bacterial artificial chromosome system (BAC) (pP-Oka) [41] and could show that it is expressed in persistently infected neuronal cells [71,72]. To visualize productively infected cells, we fused eGFP to the C-terminus of ORF11 (UL47), a tegument protein that is only expressed during lytic replication, using two-step Red-mediated *en passant* mutagenesis [73,74], Recombinant BAC clones were confirmed by PCR, DNA sequencing and RFLP using different restriction enzymes to ensure integrity of the virus genome. BAC-derived pOka strain VZV expressing GFP instead of ORF57 (pOka-Δ57-GFP) was previously described (Jacobsen et al., 2024 Nature Communications [43]). The recombinant viruses were reconstituted by transfection of BAC DNA into MeWo cells as described previously [41,75]. BAC-derived KSHV (KSHV BAC16 was previously described (Brulois et al., 2012, JV, https://doi.org/10.1128/JVI.01019-12 [42])

### Cell-free VZV preparation

Monolayers of ARPE19 cells growing in P150 dishes were used to prepare cell-free virus. ARPE19 cells were infected with the VZV cell debris or cell-associated virus. Cell-free virus was prepared when about 80% of cells were RFP positive. The infected ARPE19 cells were washed with ice-cold PBS and then detached by scraping in ice-cold PSGC (PBS containing 5% sucrose (Roth), 0.1% monosodium-glutamate (Sigma) and 10% FCS) buffer (5 ml PSGC buffer/ P150 dish). The cells were transferred into 50 ml tubes and sonicated on ice 3 times for 15 seconds with a 15 second interval with a Bandelin Sonorex RK100 sonicator. Then, the cells were centrifuged for 15 minutes at 1000 *g* at 4 °C. The supernatant was transferred to a new 50 ml tube and mixed with ice-cold Lenti-X concentrator (ratio Lenti-X:supernatant = 1:4 or 1:3). The mixture was incubated at 4 °C for 2–3 hours, followed by centrifugation at 1,500 *g* at 4 °C for 45 min and removal of the supernatant. The pellet containing 10-fold concentrated cell-free virus was resuspended and aliquoted in ice-cold PSGC buffer and stored at −80 °C.

### Titration of VZV

The determination of 50% tissue culture infection dose (TCID50) based on Spearman-Karber method was used to determine the virus titer. To this end, ARPE19 cells at a confluency of about 70% in 96 well plates (~ $10^4$ cells per well) were infected with serial dilutions of cell-free VZV. Cell-free VZV stocks were thawed in a 37 °C water bath and 10-fold serial dilutions were prepared in DMEM/F12 medium containing 2% FBS. For each viral dilution factor, 8 wells in a 96 well plate were inoculated with 100 μL/well and infection was assessed by RFP and GFP expression or cytopathic effect. The inoculum was maintained for 6 days, when the number

of wells containing RFP and GFP expression was counted, and the VZV titer was calculated according to the Spearman-Karber formula:

$$\log 10 \mathrm{TCID50} = -(X0\text{-}d/2 + d/n * \Sigma Xi).$$

X0= log10 of the reciprocal of the maximum dilution (minimum concentration) where all wells were infected; d = log10 of the dilution factor; n = number of replicates/ dilution; Xi = total number of virus-infected wells after X0, including X0. The final titer in plaque forming units per mL (PFU/mL) was calculated using the formula 0.69*TCID50/mL.

## Differentiation of SH-SY5Y cells

When SH-SY5Y cells reached approx. 60–70% density in a P100 dish, they were used for differentiation. Two types of differentiation media were used during 18-day differentiation. The cells were cultured in Differentiation Medium #1 (47.7 ml Nutrient Mixture F12 (DMEM F12) (Gibco), 1.3 ml Fetal bovine serum (Sigma), 0.5 ml GlutaMAX supplement (Gibco) and 0.5 ml penicillin–streptomycin (Cytogen)) with 10 µM All-trans retinoic acid (RA) during the first 10 days of differentiation. The medium was replaced every two days. At day 10, the cells were washed with PBS and incubated with 200 U/mL Collagenase Type IV (Gibco) diluted in DMEM/F-12 GlutaMAX medium at 37 °C for 5–10 min, until the axons of the neuron-like cells disappeared. The collagenase was gently removed and the edge of the plate was tapped to detach the neuron-like cells, leaving the epithelial-like cells still attached. The detached cells were rinsed with DMEM/F-12 GlutaMAX medium, transferred into a 50 mL centrifuge tube and centrifuged at 200 *g* for 5 min. The supernatant was removed and the cell pellet was resuspended in Differentiation Medium #2 (47 ml Neurobasal Medium (Gibco), 1 ml B-27 Supplement Minus AO (50X) (Gibco), 20 mM potassium chloride (Carl Roth), 0.5 ml Gluta-MAX supplement (Gibco), 0.5 ml penicillin–streptomycin (Cytogen), 1 mM Dibutyryl-cAMP (dbCAMP) (Selleckchem), 20 ng/mL recombinant human brain-derived neurotrophic factor (Peprotech) and 10 ng/mL recombinant human nerve growth factor (Peprotech) containing 10 µM RA. 100,000–150,000 or 50,000–75,000 cells/well were seeded on Matrigel-coated (0.15–0.16 mg/ml; Corning) 12- or 24-well plates, respectively. The cells were cultured in Differentiation Medium #2, which was replaced every two days. From day 18, the dSH-SY5Y cells were used for experiments.

## Virus infection and establishment of VZV repression state in dSH-SY5Y cells

dSH-SY5Y neuron-like cells at 18–20 days post-differentiation were employed in infection experiments. For acute infection, dSH-SY5Y cells were incubated with cell-free v63R/11G for 4 h at 37°C using an MOI based on the titer of virus obtained in ARPE19 cells. After 4 h, the virus inoculum was removed, the cells were carefully rinsed 3 times with PBS and Differentiation Medium #2 with 10 µM RA was added. To study VZV and KSHV repression, dSH-SY5Y cells were incubated with Differentiation Medium #2 containing 10 µM RA and 100 µM acyclovir (Acycloguanosine, ACV, Sigma) 24 hours prior to infection. Before infection, the supernatant of dSH-SY5Y containing RA and 100 µM ACV was harvested and used as conditioned medium. Neuron-like cells were infected with cell-free v63R/11G or KSHV in Differentiation Medium #2 containing 100 µM ACV. Mock-infected control dSH-SY5Y cells were incubated in Differentiation Medium #2 containing 100 µM ACV and the same volume of PSGC buffer as in the virus preparation used for infection. After 4 hours incubation, dSH-SY5Y cells were carefully rinsed 3 times with PBS and incubated with conditioned medium. Differentiation

Medium #2 containing 10 µM RA and 100 µM ACV was changed every two days. To induce virus replication, 10 µM PI3-kinase inhibitor (LY294002, Abcam), 2 µM histone deacetylase (HDAC) inhibitor suberoylanilide hydroxamic acid (SAHA) or lentivirus expressing eGFP or VLT-ORF63 were added at 8 dpi and incubated for 1–4 days.

### Generation of a lentivirus expressing VLT-ORF63

For the lentiviral RRLPPTSF-based VLT63-1 vector, the VLT63-1 cDNA sequence was inserted into the vector using *Age*I and *Bam*HI restriction sites. For virus production, $5 \times 10^6$ HEK 293T cells were seeded in a 10-cm dish the day before transfection. Transfections were performed using the calcium phosphate precipitation method with 5 µg RRLPPTSF-pVLT63-1 (or a RRLPPTSF-eGFP control vector [76]), 10 µg gag-pol, and 0.5 µg vsvg (packaging and envelope plasmids, respectively). Supernatants were collected 42 h and 48 h after transfection, passed through a 0.22 µm filter (Millipore), and stored at –80°C.

### DNA and RNA isolation, cDNA synthesis and quantitative PCR

Total DNA and RNA were isolated from cells using the AllPrep DNA/RNA Mini Kit (Qiagen) according to the manufacturer's instructions. The cDNA was synthesized using the Luna-Script RT SuperMix Kit (New England Biolabs) in 20 µL reaction containing 4 µL SuperMix (5X) and 1 µg RNA. Relative quantitative PCR (qPCR) and absolute qPCR were performed using a qTOWER³ Real-time Thermal Cycler (Analytik Jena). 2 µL template cDNA/DNA was used in a 20 µL reaction containing 10 µL Luna Universal qPCR Master Mix (New England Biolabs). The qPCR program was: 1 cycle of 95 °C hot start for 10 min and 45 cycles of 95 °C for 15 s and 60 °C for 45 s. For relative qPCR, viral mRNA was detected from VZV genes ORF4, ORF61, ORF62 and ORF68, human β-actin measured for normalization. For absolute qPCR, VZV ORF63 served as the viral genome target, while human β-actin served as the host genome target for normalization. PCR products were cloned into pGEM-T Easy Vector (Promega) and standard curves were generated using 10-fold serial dilutions ($10^2 - 10^9$) of templates. The copy number of the target gene in the sample was calculated by normalizing to the standard curve. Primer sequences can be found in Tables 1 and 2.

### Western blotting

Cells were lysed using radioimmunoprecipitation assay (RIPA) buffer (Sigma-Aldrich) with Pierce protease inhibitor minitablets (Thermo Fisher Scientific). Lysates were rocked for 20 min at 4 °C and centrifuged at 13,000 rpm at 4 °C for 15 min. Supernatant was collected as total protein. Stain-free total protein detection was performed as previously described [77]. Briefly, protein samples were mixed with SDS loading buffer, heated at 98 °C for 5 min and

**Table 1.  Sequences of primers used to quantify gene expression.**

| Name | Sequence (5′–3′) |
| --- | --- |
| ORF4_F | GCCCATGAATCACCCTC |
| ORF4_R | ACTCGGTACGCCATTTAG |
| ORF61_F | GGACAGACTGCCTTTTCGAG |
| ORF61_R | GACAACGCAGGGATGTTTTT |
| ORF68_F | GTACATTTGGAACATGCGCG |
| ORF68_R | TCCACATATGAAACTCAGCCC |
| actin_F | TCATCACCATTGGCATGAG |
| actin_R | AGCACTGTGTTGGCGTACAG |

**Table 2. Sequences of primers used to determine genome copy number.**

| Name | Sequence (5′–3′) |
| --- | --- |
| VZV-ORF63_F | CCCGGCGCGTTTTGTACTCC |
| VZV-ORF63_R | ACAATTCCTCCCAGCACGCTA |
| h-β-Actin_F | TCCTCCTGAGCGCAAGTACTCC |
| h-β-Actin_R | AAGTCATAGTCCGCCTAGAAGCA |

loaded into SDS-PAGE gels containing 1% 2,2,2-Trichloroethanol (TCE). Total proteins were visualized by illumination with UV light using ChemiDoc MP Imaging System (Bio-Rad). The separated proteins were transferred onto nitrocellulose membranes and then blocked in 5% skimmed milk plus PBS-0.1% Tween 20 (PBS-T). Primary antibodies were diluted in PBS-T containing 5% skimmed milk and incubated overnight at 4 °C. Membranes were then washed 3 times with PBS-T buffer for 10 min and then incubated in PBS-T containing 5% skimmed milk and fluorescently-conjugated secondary antibody for 1 h at room temperature. Membranes were then washed as described above and detection was performed with ChemiDoc MP Imaging System (Bio-Rad). The antibodies were as follows: mouse monoclonal anti-VZV gE (LSBio Biozol, 1:2,000); mouse anti-VZV ORF4 (CapRi, 1:1,000); mouse monoclonal anti-VZV ORF63 Cl.63.08, kappa IgG1 (Capri Center for Proteomics, 1:1,000); mouse monoclonal anti-β-actin (Thermo Scientific, 1:5000); mouse monoclonal anti-RFP-antibody (3F5) (ChromoTek GmbH, 1:1000); mouse monoclonal anti-GFP (TaKaRa, 1:1000); anti-mouse IgG IRDye 800 (LI-COR, 1:10,000).

## Immunofluorescence

Cells were washed with PBS, fixed for 20 min with 4% paraformaldehyde at room temperature, washed again and incubated in permeabilizing and blocking solution (0.5% Triton X-100, 5% BSA) for 1 h. Cells were labelled with the following primary antibodies diluted in PBS containing 0.1% Triton X-100, 1% BSA at 4 °C overnight: rabbit anti-microtubule-associated protein 2 (MAP2, Millipore 1: 200); sheep polyclonal anti-dopamine beta hydroxylase (DBH, Thermo Fisher Scientific, 1:50); Tuj1 mouse anti-tubulin-β-III (Tuj1, Millipore, 1:300); rabbit polyclonal anti-Nav 1.7 (Alomone, 1:200); rabbit polyclonal anti-KI67 (Proteintech, 1:400). Cells were then washed, incubated 1 h at room temperature with DAPI and conjugated secondary antibodies: anti-mouse IgG Alexa Fluor 488 (Life Technologies, 1:1000); anti-mouse IgG Alexa Fluor 555 (Life Technologies, 1:1000); anti-sheep IgG Alexa Fluor 488 (Life Technologies, 1:1000). Cells were washed and mounted onto glass slides with Prolong Gold Antifade Mountant (Thermo Fisher). Images were obtained using a Zeiss observer Z1 inverted microscope.

## Chromatin Immunoprecipitation (ChIP)

ChIP was performed as described previously [47]. Cells were cross-linked (1% formaldehyde, 10 minutes), quenched with 125 mM glycine, washed twice with PBS and harvested in 1 ml buffer 1 (50 mM Hepes-KOH, 140 mM NaCl, 1 mM EDTA, 10% glycerol, 0.5% NP-40, 0.25% Triton X-100) and incubated for 10 min at 4 °C while rotating. After centrifugation (1,350 x g, 5 min), nuclei were incubated with 1 ml buffer 2 (10 mM Tris-HCl, 200 mM NaCl, 1 mM EDTA, 0.5 mM EGTA) for 10 min at 4 °C while rotating. Pelleted nuclei were lysed in buffer 3 (1% SDS, 10 mM EDTA, 50 mM Tris-HCl). Chromatin was sonicated (fragment size 200–500 bp) using a Bioruptor (Diagenode). After addition of Triton X-100 (1% final concentration) cell debris was pelleted (20,000 x g, 4°C) and chromatin containing supernatant

was collected. Chromatin of $1x10^6$ cells was diluted 1:10 in dilution buffer (0.01% SDS, 1.1% Triton X-100, 1.2 mM EDTA, 16.7 mM Tris-HCl, 167 mM NaCl) and incubated with respective antibodies overnight. 50 µl BSA-blocked Protein A/G Magnetic Beads (Pierce) was added to precipitate the chromatin-immunocomplexes and incubated for 3 hr at 4°C. Beads were washed once with 1 ml of the following buffers: low-salt buffer (0.1% SDS, 1% Triton X-100, 2 mM EDTA, 20 mM Tris-HCl, 150 mM NaCl); high-salt buffer (0.1% SDS, 1% Triton X-100, 2 mM EDTA, 20 mM Tris-HCl, 500 mM NaCl); LiCl-wash buffer (0.25 M LiCl, 1% Nonidet P-40, 1% Na-deoxycholate, 1 mM EDTA, 10 mM Tris-HCl) and TE-wash buffer. Chromatin was eluted and decrosslinked from the beads by incubation in 120 µl SDS containing elution-buffer (50 mM Tris-HCl pH 8.0, 10 mM EDTA, 1% SDS) containing 200 mM NaCl at 65 °C overnight. Chromatin containing supernatant was separated from the beads by a magnetic rack. DNA was purified using a DNA Clean & Concentrator kit (Zymo Research). For ChIP-seq, 1–2 ng of ChIP DNA was used for library preparation, using the NEBNext Ultra II DNA Library prep Kit (E7370; NEB). Libraries were sequenced using an Illumina NextSeq 2000 sequencer 75 bp Single End.

## Assay for Transposase-Accessible Chromatin using sequencing (ATAC-seq)

ATACseq was performed using the Omni-ATAC-seq protocol [78]. Briefly, $1x10^5$ cells were treated with DNase I (200 U/ml, Worthington) at 37°C for 30 min, washed with cold PBS twice and resuspended in 1 ml cold RSB buffer (10 mM Tris-HCl pH 7.4, 10 mM NaCl, 3 mM $MgCl_2$). Cells were pelleted again at 500 x g for 5 min and resuspended in 50 µl of cold ATAC-NTD lysis buffer (RSB Buffer + 0.1% NP40, 0.1% Tween-20, 0.01% Digitonin). Lysed cells were diluted in 1 ml cold ATAC-T buffer (RSB + 0.1% Tween-20) and inverted three times. The resulting nuclei were pelleted at 500 x g for 10 minutes and the supernatant was removed. Cell pellets were transposed with 50 µl of transposition mix containing 25 µl 2xTD Buffer (20 mM 1M Tris-HCl pH 7.6, 10 mM $MgCl_2$, 20% Dimethyl Formamide), 2.5 µl transposase (custom made, 100 nM final), 16.5 µl PBS, 0.5 µl 1% digitonin, 0.5 µl 10% Tween-20 and 5 µl $H_2O$) at 37°C and 1000 rpm on a thermomixer for 30 min. The reaction was stopped by adding 250 µl of DNA Binding Buffer and DNA was isolated using the Clean and Concentrator-5 Kit (Zymo). Libraries were produced by PCR amplification of tagmented DNA and sequenced on a NextSeq 2000 sequencer 150 bp Paired End. For fragment size estimation libraries were resequenced on a NextSeq 2000 sequencer 50 bp Paired End to achieve >100 million reads per sample.

## Sequencing data processing

For ChIP-seq, quality filtered single end reads were aligned to the viral reference genome of VZV (NC_001348.1), KSHV (HQ404500) and human (hg38) using Bowtie [79] with standard settings. Analysis of histone modification enrichment on the VZV genome was done as described in [80]. Briefly, host positive sites were defined using peak finding via MACS2 [81] (H3K4me3, H3K27ac) or EPIC2 [82] (H3K27me3, H2AK119Ub, H3K9me3 and H3). The 200 most significant enriched regions were selected as host positive regions. Negative regions, which represent the general ChIP background of each individual analysis were generated by random selection of regions with the same size distribution as the positive controls. We excluded regions with less than 10 reads per region in the input as well as all regions with known mappability bias (blacklists downloaded from ENCODE). Viral sequences were split into segments of 10 kb (shifted by 5 kb) to reflect broad regions. We counted the reads within each positive, negative and viral region using FeatureCounts [83]. For each individual region we then calculated the ChIP-to-input read count ratio and normalized all groups to the

median of the respective negative control. The resulting values represent relative enrichment of the respective histone modification signals over the general ChIP background present in each individual experiment. Significance of enrichment of positive controls and viral regions over negative (background) controls was calculated via Wilcoxon-Mann-Whitney-Test.

For ATACseq, reads were processed using the PEPATAC pipeline [84]. Exact integration sites were extended +/-25 bp and visualized using IGV tools. Global analysis of ATAC-seq signal enrichment on the VZV genome was done similarly as described above for ChIP-seq. Enriched regions were identified using MACS2, signal was quantified on all positive regions (n=54760) and compared to a size-matched collection of randomly selected control regions (n=54760) and 500 bp sliding windows across the VZV genome (n=249). Average read density of ChIP-seq and ATACseq data at human transcriptional start sites (+/−2.5 kbp) was calculated and visualized using EaSeq [85]. To estimate nucleosome periodicity, fragment insert size of ATAC-seq libraries sequenced at >100 million reads were analyzed using Picard CollectInsertSizeMetrics (Picard toolkit, 2019). To compare nucleosome periodicity, host (hg38) aligned reads were randomly subsampled to the number of total aligned reads to VZV (NC_001348.1) using sambamba [86].

## RNA in situ hybridization (RNAscope)

RNAscope was performed using the RNAscope Fluorescent Multiplex Kit (ACD BioTechne). In brief, cells on 8-well chambers were fixed with 4% PFA for 20 min at RT and overnight at 4 °C. On the next day, the cells were incubated with RNAscope hydrogen peroxide for 10 min at RT followed by protease digestion for 10 min at RT. After washing with PBS, the cells were incubated with pre-mixed target probe (RNAscope Probe-V-VZV-O2, targeting VZV VLT) or control probes (RNAscope 3-Plex Negative control Probe/ RNAscope 3-Plex Positive control Probe_Hs), both designed by ACD BioTechne, for 2 hours at 40°C in the HybEZ hybridization oven (ACD). Cells were washed with 1x wash buffer and incubated with amplification reagents (Amp 1 for 30 min, Amp 2 for 30 min and Amp 3 for 15 min at 40°C). After washing with 1x wash buffer, the cells were incubated with HRP adaptor for 15 min at 40°C, followed by incubation with the corresponding dye for 30 min at 40°C and incubated with HRP blocker for 15 min at 40°C. Cells were counterstained with DAPI and mounted onto glass slides with Prolong Gold Antifade Mountant (Thermo Fisher). Images were obtained using a Zeiss observer Z1 inverted microscope and Leica Inverted-3 microscope and analyzed by Fiji.

## DNA in situ hybridization (DNAscope)

Viral DNA detection was performed using the RNAscope Fluorescent Multiplex Kit (ACD BioTechne) with modifications using the RNAscope Probe-V-VZV-O2, targeting VZV VLT, designed by ACD BioTechne. Briefly, we performed an RNase treatment with Resuspension buffer A1 containing RNase and 0.05% Tween-20 for 30 min at 40°C after the protease digestion step. The negative control wells were incubated with DNaseI for 40 min at 40°C after RNase treatment and washed 3 times with PBS containing 1 mM EDTA to inactivate DNaseI. We also performed a short denaturation step by incubating the 8-well chamber at 60°C with pre-warmed (60°C) probe for 10 min, and then immediately transferred the chamber to the oven at 40°C, followed by hybridization overnight. Amplification and detection were performed as described for RNAscope (see above), using 0.5x wash buffer for all washing steps.

## RNA-Seq library preparation and sequencing

For each sample, polyadenylated (poly(A)) RNA was isolated from one microgram of total RNA using the NEBNext Poly(A) mRNA Magnetic Isolation Module. Reverse transcription,

second strand synthesis, end-repair and A-tailing were subsequently performed using the NEBNext Ultra II Directional RNA Library prep kit. For the adaptor ligation step, we used TWIST Universal Adapters from the standard TWIST Library Preparation Kit and omitted the addition of NEB USER enzyme. Resulting libraries were subsequently amplified (six cycles of PCR) using TWIST UDI primers and the Equinox Library Amp Mix, all according to the protocols laid out in the TWIST Library Preparation manual. Resulting libraries were purified using AMPure XP beads and subsequently multiplexed in equimolar ratios. Hybridization was performed for 18 hours using biotinylated oligos designed by TWIST Biosciences against all known VZV genome sequences. Post-hybridization washes and amplification (18 cycles of PCR) were used to produce the final multiplexed library which was subsequently sequenced on an Illumina MiSeq using a 2x150 bp Micro Kit.

### RNA-Seq analysis

Sequence data were de-multiplexed and individual sequence data sets were trimmed using the TrimGalore software (http://www.bioinformatics.babraham.ac.uk/projects/trim_galore/) to remove adaptor sequences and low-quality 3' ends. Sequence reads were competitively aligned against the human (HG38) and VZV genomes (strain Dumas, NC_001348) using STAR v2.7.9 (https://www.ncbi.nlm.nih.gov/pmc/articles/PMC3530905/). De-duplication of aligned reads was performed using picardtools MarkDuplicates (http://broadinstitute.github.io/picard). Resulting assemblies were parsed using SAMTools v1.15 (https://pubmed.ncbi.nlm.nih.gov/19505943/) and BEDTools v2.27 (https://pubmed.ncbi.nlm.nih.gov/20110278/) to produce bedgraphs that were visualized in Rstudio using the package GVIZ (https://pubmed.ncbi.nlm.nih.gov/27008022/).

### Supporting information

**S1 Fig. Incubation with ACV progressively represses VZV Δ57-GFP in dSH-SY5Y cells.** Relative gene expression of VZV genes, n=3 (A-C) and quantification of VZV genomes (D), n=3, in VZV Δ57-GFP infected dSH-SY5Y cells in the absence (acute) or presence of ACV for 6 dpi. Statistical comparisons between acutely infected cells and rest of the conditions were performed using one-way ANOVA on log-transformed raw data to stabilize variances and improve normality. $P > 0.05$ (ns), $P \leq 0.05$ (*), $P \leq 0.01$ (**), $P \leq 0.001$ (***), $P \leq 0.0001$ (****). Abbreviations: ACV, acyclovir
(TIF)

**S2 Fig. Expression of pVLT-ORF63 and eGFP in ARPE-19 cells transduced with the VLT-ORF63 and eGFP lentiviral supernatants.** Western blot showing expression of pVLT-ORF63 (top) and eGFP (bottom) in lysates of ARPE19 cells transduced with the respective lentivirus. Abbreviations: kDa, kilo Daltons.
(TIF)

**S3 Fig. Ectopic pVLT-ORF63 expression induces VZV protein expression in infected dSH-SY5Y cells incubated with ACV for 6 days.** Western blots showing VZV proteins after incubation with the indicated antibodies and total protein detected with TCE staining (bottom blots) in lysates of dSH-SY5Y cells mock-infected or infected with v63R/11G in the absence (acute) or presence of ACV for 6 days and transduced with a lentivirus expressing eGFP or pVLT-ORF63. NR refers to "non-replicating" VZV, while R refers to "replicating" VZV, based on the lack or presence of ORF63-RFP/ORF11-GFP expression. Abbreviations: kDa, kilo Daltons; ACV, acyclovir.
(TIF)

**S4 Fig. Detection of VLT-associated transcripts in dSH-SY5Y cells incubated with ACV and infected with VZV.** (**A,B**) Expanded views of transcription profiles in the VLT and ORF63 regions of dSH-SY5Y cells infected with v63R/11G in the absence (acute, violet) and presence of ACV for 6 dpi (orange, A; blue, B) or for 3, 4, 5 and 6 dpi in the presence of ACV with samples analyzed 2 days after ACV removal (orange, B). Bulk RNA-Seq was performed at different times post-infection, as labelled. Transcription from both DNA strands is shown with the depth of coverage labelled on the y-axis. A representation of the VLT and ORF63 regions is shown. Abbreviations: dpi, days post-infection; ACV, acyclovir. (TIF)

**S5 Fig. ChIP-seq and ATAC-seq analysis of differentiated SH-SY5Y cells quiescently infected with VZV and KSHV at 13 days post-infection.** (**A**) Average read density of ChIP-seq and ATAC-seq reads at all human TSS (+/−2.5 kb) from dSH-SY5Y cells quiescently infected with VZV (VZV pOKA WT) at day 13 p.i. (**B**) Input-normalized quantification of ATAC-seq coverage at all positive host sites (n=54760) compared to a count/size-matched collection of randomly selected control regions and 500 bp sliding windows across the VZV genome (n=249) from dSH-SY5Y cells quiescently infected with VZV (VZV pOKA WT) at day 13 p.i. (**C**) ChIP seq data of dSH-SY5Y cells latently infected with KSHV (BAC16) at day 13 p.i. Input-normalized quantification of ChIP-seq signals in a 10 kb sliding window across the KSHV genome (KSHV BAC16, right panel), relative to the 200 most significantly enriched host regions (hg38 positives, left panel) and an equal number of size matched, randomly selected host control loci (hg38 negatives, center). Signals observed in host control regions were set to 1 (10E0). (P > 0.05 (n.s), P ≤ 0.05 (*), P ≤ 0.01 (**), P ≤ 0.001 (***) Wilcoxon-Mann-Whitney-Test of indicated sample compared to corresponding hg38 negatives). (**D**) Read density coverage tracks of histone marks on two host loci and KSHV, determined by ChIP-seq. (**E**) Number of read alignments of ChIP-input samples to human (hg38) and virus (VZV pOKA WT or KSHV BAC16). Number of reads were normalized to genome size (hg38 = 3.049 Gb, VZV = 124 kb, KSHV = 136 kb) and sequencing depth as RPKM values to estimate the fold enrichment of viral over host as an approximation of viral copy numbers per cell. Abbreviations: FPKM, fragments per kilobase per million mapped fragments; pos, positive; neg, negative; chr, chromosome; RPKM, reads per kilobase per million mapped reads. (TIF)

**S1 Table. Quantification of DNAscope results obtained at 12 dpi in dSH-SY5Y cells infected with VZV and incubated with ACV during 5 days.** The table summarizes the number of replicates, fields of view analyzed, VLT DNA positive cells, and total of DAPI-stained cells. The percentage of VLT DNA positive cells (5.1%) represents the proportion of VLT DNA positive cells within the total DAPI positive cells. (DOCX)

**S2 Table. Quantification of DNAscope results obtained at 20 dpi in dSH-SY5Y cells infected with VZV and incubated with ACV during 5 days.** The table summarizes the number of replicates, fields of view analyzed, VLT DNA positive cells, and total of DAPI-stained cells. The percentage of VLT DNA positive cells (4.9%) represents the proportion of VLT DNA positive cells within the total DAPI positive cells. (DOCX)

**S3 Table. Quantification of DNAscope results obtained at 12 dpi in dSH-SY5Y cells infected with VZV and incubated with ACV during 6 days.** The table summarizes the number of replicates, fields of view analyzed, VLT DNA positive cells, and total of DAPI-stained

cells. The percentage of VLT DNA positive cells (4.4%) represents the proportion of VLT DNA positive cells within the total DAPI positive cells.
(DOCX)

**S4 Table. Quantification of DNAscope results obtained at 20 dpi in dSH-SY5Y cells infected with VZV and incubated with ACV during 6 days.** The table summarizes the number of replicates, fields of view analyzed, VLT DNA positive cells, and total of DAPI-stained cells. The percentage of VLT DNA positive cells (4.4%) represents the proportion of VLT DNA positive cells within the total DAPI positive cells.
(DOCX)

**S5 Table. Quantification of RNAscope results obtained at 12 dpi in dSH-SY5Y cells infected with VZV and incubated with ACV during 6 days.** The table summarizes the number of replicates, fields of view analyzed, VLT RNA positive cells, and total of DAPI stained cells. The percentage of VLT RNA positive cells (1.0%) represents the proportion of VLT RNA positive cells within the total DAPI positive cells.
(DOCX)

**S6 Table. Quantification of RNAscope results obtained at 20 dpi in dSH-SY5Y cells infected with VZV and incubated with ACV during 6 days.** The table summarizes the number of replicates, fields of view analyzed, VLT RNA positive cells, and total of DAPI stained cells. The percentage of VLT RNA positive cells (0.9%) represents the proportion of VLT RNA positive cells within the total DAPI positive cells.
(DOCX)

## Acknowledgments

We thank Paul (Kip) R. Kinchington (University of Pittsburgh, USA) for providing the pOka-Δ57-GFP strain, Nikolaus Osterrieder (University of Veterinary Medicine Hannover, Germany) and Cornell University (New York, USA) for the wt BAC-derived VZV pOka strain, Werner Ouwendijk and Georges Verjans (Erasmus MC, Rotterdam, The Netherlands) for the original protocol to produce cell-free VZV, Tanvi Tikla and Thomas F. Schulz (Institute of Virology, Hannover Medical School, Germany) for the KHSV BAC36 and Jens Bohne (Institute of Virology, Hannover Medical School, Germany) for providing the lentivirus expressing eGFP..

## Author contributions

**Conceptualization:** Jiayi Wang, Adam Grundhoff, Abel Viejo-Borbolla.

**Data curation:** Simon Weissmann, Lars Steinbrück, Dan P. Depledge.

**Formal analysis:** Jiayi Wang, Simon Weissmann, Thomas Günther, Dan P. Depledge.

**Funding acquisition:** Adam Grundhoff, Abel Viejo-Borbolla.

**Investigation:** Jiayi Wang, Nadine Brückner, Simon Weissmann, Shuyong Zhu, Carolin Vogt, Guorong Sun, Rongrong Guo, Renzo Bruno, Birgit Ritter, Lars Steinbrück, Dan P. Depledge.

**Methodology:** Jiayi Wang, Thomas Günther, Shuyong Zhu, Renzo Bruno.

**Project administration:** Abel Viejo-Borbolla.

**Resources:** Benedikt B. Kaufer, Abel Viejo-Borbolla.

**Supervision:** Daniel P Depledge, Adam Grundhoff, Abel Viejo-Borbolla.

**Validation:** Nadine Brückner, Simon Weissmann.

**Writing – original draft:** Jiayi Wang, Abel Viejo-Borbolla.

**Writing – review & editing:** Jiayi Wang, Nadine Brückner, Simon Weissmann, Carolin Vogt, Benedikt B. Kaufer, Daniel P Depledge, Adam Grundhoff, Abel Viejo-Borbolla.

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
