## [Decision Letter · Decision Letter 0]

19 Jul 2024

Dear Professor Viejo-Borbolla,

Thank you very much for submitting your manuscript "Repression of varicella zoster virus gene expression during quiescent infection in the absence of detectable histone deposition" for consideration at PLOS Pathogens. As with all papers reviewed by the journal, your manuscript was reviewed by members of the editorial board and by several independent reviewers. In light of the reviews (below this email), we would like to invite the resubmission of a significantly-revised version that takes into account the reviewers' comments.

All three reviewers found this work to be compelling, but agreed that additional characterization regarding the model was warranted. It is important to address each of the comments provided. In addition, the authors should carefully quantitate by multiple methods (qPCR and GFP expression) genomes in the quiescent cells. The reviewers all agree that there are alternative explanations for the lack of reactivation observed in the model, including no infection (or very low level of infection with VZV) and additional experimentation is warranted here to more conclusively support the conclusions presented. The authors should also provide statistical analyses for the data presented and consider performing microscopy experiments that show whether the quiescent genomes are trapped in PML bodies or capsids, two interesting hypotheses brought up in the future directions and discussion.

We cannot make any decision about publication until we have seen the revised manuscript and your response to the reviewers' comments. Your revised manuscript is also likely to be sent to reviewers for further evaluation.

Sincerely,

Donna M Neumann

Academic Editor

PLOS Pathogens

Robert Kalejta

Section Editor

PLOS Pathogens

Michael Malim

Editor-in-Chief

PLOS Pathogens

orcid.org/0000-0002-7699-2064

All three reviewers found this work to be compelling, but agreed that additional characterization regarding the model was warranted. It is important to address each of the comments provided. In addition, the authors should carefully quantitate by multiple methods (qPCR and GFP expression) genomes in the quiescent cells. The reviewers all agree that there are alternative explanations for the lack of reactivation observed in the model, including no infection (or very low level of infection with VZV) and additional experimentation is warranted here to more conclusively support the conclusions presented. The authors should also provide statistical analyses for the data presented and consider performing microscopy experiments that show whether the quiescent genomes are trapped in PML bodies or capsids, two interesting hypotheses brought up in the future directions and discussion.

Reviewer's Responses to Questions

**Part I - Summary**

Reviewer #1: In this manuscript, Wang et al characterize a novel neuronal cell line based system for the study of VZV latency/quiescence. They discover the conditions necessary for differentiated SH-SY5Y to harbor VZV genomes in the absence of viral replication and near absence of viral transcription. Ultimately, these cells can be reactivated by direct addition of viral transcription activating proteins. Overall I think this is a well controlled paper documenting an interesting system and very interesting findings. I have only minor comments that I feel would improve the clarity and presentation of the findings.

Reviewer #2: In their study, Wang et al. explore the potential of the neuroblastoma-derived cell line SH-SY5Y as a model to investigate the mechanisms involved in the repression and subsequent reactivation of VZV gene expression. They demonstrate that differentiated SH-SY5Y (dSH-SY5Y) cells support productive infection by VZV. Treatment with acyclovir (ACV) inhibits viral replication and leads to a gradual repression of viral activity. After ACV removal, some cells produce viral particles, while others harbor non-replicating VZV genomes and transcripts containing VLT for at least 20 days post-infection (dpi). Introduction of exogenous VLT-ORF63 induces productive infection, indicating that the repressed genomes in non-replicating state remain capable of functionality. Histone deposition is notably absent at VZV genomes in quiescently infected dSH-SY5Y cells, suggesting a potentially novel mechanism of VZV repression in this neuronal context.

Developing neuronal models to study neurotropic herpesviruses is essential for understanding their biology, particularly latency and reactivation processes. In this context, the study by Wang et al. thoroughly explores the use of dSH-SY5Y cells as a potential additional cell culture model for investigating VZV latency. However, a significant limitation in this reviewer's opinion is the lack of a deeper investigation into the ACV model of VZV latency establishment. Specifically, the failure to achieve reactivation with LY294002 or SAHA does not align with previous studies by Markus et al. (PLoS Path, 2015). A more significant concern is the inability to detect histones and modified histone marks on the quiescent VZV in this model, despite these being described by Gary et al. (JVI, 2006) using human TG. The reviewer has confidence in the authors' proficiency in performing ChIP experiments. However, without using additional viral tools to establish a VZV latency-like state in dSH-SY5Y cells and conducting further ChIP, CUT&Run, or CUT&Tag analyses, it remains unclear whether the observed putative non-nucleosomal form of latent VZV is an intrinsic property of the virus or a bias introduced by using ACV. Consequently, there are major questions regarding the accurate use of this model. Without deeper mechanistic analyses, the study serves as a strong description of a cell culture model for investigating VZV latency but likely falls short in providing insightful answers to key questions about VZV biology.

Reviewer #3: Wang et al. present an interesting study investigating Varicella zoster virus (VZV) latency in human neurons. They develop a cost-effective model by modifying the neuronal differentiation protocol of the SH-SY5Y cell line to produce a predominantly neuronal-like cell type (dSH-SY5Y). Importantly, these cells can be infected with VZV and treatment with acyclovir (ACV) for 6 days post infection (dpi) can prevent induction of VZV genes while retaining low levels of VZV genomes within the cells. Interestingly, treatment with ACV for 5 dpi establishes a heterogeneous population of cells with around half exhibiting VZV replication and half exhibiting no VZV replication. This condition could be effective for studying how VZV latency is induced and controlled from both the host and virus side. Nevertheless, the authors focused on the ACV 6dpi treatment. The authors then use RNA-seq, ChIP-seq, and ATAC-seq to conclude that there are quiescent VZV genomes that are free of histone deposition. While the premise is interesting, the study appears to be lacking statistical power that would be critical to evaluating their data. Similarly, oftentimes the authors overinterpret some of their results, specifically as relates to histones, and as such unfortunately disregard the underlying biology, raising questions about the impact of this study.

**Part II – Major Issues: Key Experiments Required for Acceptance**

Reviewer #1: 1. For all images scale bars could be drawn thicker and larger. It would help to show the scale of the clusters of neural-like nuclei.

2. In Figures 2 and 3 it is said the presence of both ORF63-RFP and ORF11-GFP denotes active replication. Especially in the ACV withdrawal experiments in Figure 3 are there ever any cells that are just ORF63-RFP positive which might be indicative of an early reactivation without progression to the late stage?

3. Often the total number of experiments is displayed, but there is no mention of biological replicates that might be controlling for the batch effects with the 20-day neuronal differentiation culture experiments. In essence, in Figure 3 were the 6 dpi n=70 wells done with one batch differentiated neurons or accumulated over multiple independent experiments. Same with the 2 dpi n=8, etc. There are also multiple instance in the manuscript where the percentage of cells displaying a phenotype is mentioned in the text, but there is no accompanying figure that could also give key breakdowns of experiment to experiment variability. This should be rectified for Figure 4G (5% of cells genome positive, over how many biological replicates/fields of view), Figure 5H (4.4% of cells genome positive, at 12 dpi and 20 dpi?), and Figure 7C (% of cells with VLT RNA versus genome containing).

4. For the ChIP-Seq experiments it is stated that all experiments were done with both WT and v63R/11G viruses, however there are only tracks shown for one replicate and Figure 8C is simply labeled Replicate 1 and 2. Please explain.

5. The lack of any appreciable histone signal is a bold claim that appears to be supported by the internal matching cellular data. One additional piece of evidence supporting this claim might also already exist within the paired end ATAC-seq data: If the ATAC insert size between the two paired end reads is mapped you often get a periodicity of reads in multiples of ~160bp, i.e., the size of DNA wrapped around a nucleosome. Does the ATAC data for the human genome have this periodicity while the viral genome does not?

Reviewer #2: Here are some suggestions that could potentially enhance the content of the manuscript.

Major:

1) Reactivation experiments using LY294002 and SAHA were supposedly conducted at 34°C, though this is not specified in the Methods section. According to Markus et al., PLoS Path, 2015, which the authors reference, LY294002 appears more effective at 34°C for reactivating VZV in latently infected ESC-derived neurons. The authors should attempt these conditions before concluding the inefficacy of LY294002 and SAHA.

2) Suppl Fig3: The "NR" samples express lower levels of pVLT-ORF63. Could their failure to reactivate be due to the ectopic pVLT-ORF63 levels being below a necessary threshold? How to address this issue? If a threshold aspect is to be considered, then the concluding sentence: "These results show that pVLT-ORF63 releases the repression on the VZV genome in dSH-SY5Y cells incubated with ACV for 6 days, highlighting that the infection was not abortive" should be revised to incorporate the potential influence of a pVLT-ORF63 threshold

3) It is difficult to believe that a DNA entity could remain in the nucleus in a non-nucleosomal form. However, this is what the data obtained by ChIP-seq and ATAC-seq appear to indicate. To my knowledge, the chromatinization state of VZV during infection is not well understood. However, Gary et al. (JVI, 2006) reported that the H3K9ac chromatin mark is associated with latent VZV genomes, at least on ORF62 and ORF63 in human TG, and on ORF62, ORF63, ORF36, and ORF14 during productive infection of MeWo cells. The investigation of this mark is missing in the Wang et al. study and should be included.

Additionally, the results obtained by Wang et al. could be attributed to the latency model using ACV to artificially block viral replication and induce a latency-like state. However, numerous studies using ACV to establish an HSV latency-like state in cell culture models indicate that ACV does not prevent HSV chromatinization. However, the situation could be different for VZV. Figure 4 illustrates that early after infection in the presence of ACV, viral products such as ORF61 and ORF68 are expressed. These viral products or others present or not in the viral particle tegument could potentially impact viral chromatinization by influencing cellular chromatinization mechanisms in some way. If this is indeed the case, VZV would have evolved a very interesting putative mechanism to evade chromatinization and potentially avoid repression through this process. However, findings from Gary et al. suggest that such an assumption may not align with their observations. It is crucial to clarify this aspect by employing alternative methods to establish a latency-like state, such as using a replication-defective VZV mutated in an essential immediate-early gene, or UV-inactivated virus capable of infecting SH-SY5Y cells. Without such alternative approaches to study VZV latency, readers might perceive the absence of chromatinization of quiescent VZV under ACV as a limitation of the entire study. Significantly, the lack of reactivation observed with LY294002 and SAHA as depicted in Figure 6 could be attributed to the absence of a chromatinized state in this model. The low percentage of cells positive for VZV DNA, as depicted by FISH in Fig. 5H, may also be associated with the ACV-induced quiescent state of VZV and the reduction in viral genomes per cell, as measured in Figures 4F and 5G.

Reviewer #3: 1) The lack of viral genomes does not necessarily indicate no reactivation, it may just mean that there is no infection to begin with. It is unclear whether the ‘clearing’ of the viral genomes means they have latent infection or they are simply not infected if no viral genomes can be detected. Latent viral genomes are detectable in many systems (HPV, KSHV, EBV etc), which would be an important aspect of a functional latency model. Also, it is difficult to compare the findings between groups. For example in figure 3, genomes present in these wells measured by qPCR would be more compelling than simply not RFP/GFP positive, and from there a ratio of which wells have detectable VZV genomes to RFP/GFP would enable validation of virus in the wells and a more convincing latency model. Also figure 4B and 5B would need the VZV genome staining to confirm the cells displayed are latently infected and not just uninfected.

2) The conclusions based on the number of viral genomes detected are highly problematic. There are several reasons for this concern: 1) they detect viral genomes in only 5% of the cells, which is more indicative of no infection that latency, as stated above; 2) The conclusion that there are no histones on the viral genomes based on their findings is not convincing. The lack of statistical analysis also suggests that these datasets were not thoroughly analyzed, especially when the data presented indicate that there are peaks of histones (eg H3K27me3 and H3K9me3) that with different analyses and greater signal would be apparent. Given the biology that all known latent viruses have repressive chromatin deposited, it is a higher burden of proof to say the opposite, which would require more systematic approaches and analysis than just lack of detection. For example, if the authors had a higher percentage of detectable latent genomes, then statistical power would support their conclusions. However, with such low numbers, lack of signal is not the same as saying histones are not present. One possible experiment could be to stain for genomes by FISH in a cell that expresses an H3-GFP or similar tagged histones to look for co-localization, as has been done in other systems; 3) If the conclusion that histones are not present were true, then the authors should suggest a different model for repression or lack of gene expression. It has been well established that open chromatin or nucleosome-free regions are highly expressed, not silent; 4) Another concern in the analysis is the comparison to input and the host genome. Because the viral genome is much smaller in comparison to the host, and the input sample is much higher in concentration by definition, then it is expected that the reads would be low. A positive control for a protein known to be bound to the viral genomes would be critical here to ensure that the ChIP is an accurate measure of proteins bound to the viral genomes. The text on page 16 and figure 8c describes positive and negative controls on the host genome, but no positive control for the viral genome was used, which calls into question whether the ChIP of viral genomes was successful enough to be interpreted. Similarly, if the genome lacked any periodicity of bound proteins, the ATAC-seq signal should be uniformly high, however, there are peaks and patterns in the ATAC-seq data. Further experimentation and development of the model on viral genomes is required to draw conclusions.

3) Quantification of in situ hybridization data is included in the text but no quantification is provided in the figures. Also controls such as Mock infected and 1 or 6 dpi infection are missing.

**Part III – Minor Issues: Editorial and Data Presentation Modifications**

Reviewer #1: 1. I find Days Post Infection and Days Post release confusing when discussing the NR wells versus the R wells, as both conditions could be said to be either DPI or DPR regardless of whether they start replicating or not. Maybe this could be made more clear in Figure 4A?

Reviewer #2: 1. Fig3B: The abscissa indications of graph B are confusing and do not align with the explanations provided in the text.

2. Figure 4B: The annotation "R" with the indication "5 days ACV treatment" is confusing, as it is similar to the "R" present in panels D, E, F, which pertains to days post-release from ACV treatment.

3. Figures 4F and 5G show a decrease in the amount of VZV genomes with time up to 30 dpi. How do the authors explain this?

4. Figure 4A, 5A, 6A: The timeline lacks the indication of the days post-infection (dpi) when cells are analyzed.

5. The message conveyed by Figures 5 adds incrementally to Figure 4. Figures could be merged, or Figure 5 could be included in the supplementary figures.

6. Figures 4F, 5G: How do the authors account for the decrease in VZV genomes per cell? Is this linked to the ACV model of latency establishment? Could it be related to the absence of chromatinized quiescent genomes under this treatment?

7. Figure 5C: IE4 and other IE, E proteins staining should be examined at earlier times post-infection (pi), starting from 2 dpi, to investigate whether IE proteins are produced during the period between 2 and 6 dpi.

8. Legend Figure 7: The figure description does not fully correspond with the visual content presented.

9. Figure A: what are the samples 12, 16, 20, 30 dpi?

10. Figure B: The legend should be revised to accurately explain the representation of 5, 6, 7, and 8 dpi in orange, as it currently refers to 3, 4, 5, and 6 dpi. The legend should clarify the timescale of ACV addition/removal.

Typos

P3l65: change ‘’infected cell polypeptide 0’’ by ‘’ ’infected cell protein 0’

P55l1145-47, Figure 2 legend: change (B) by (C); (C-E) by (D-F); and (B-E) by (C-F)

P9l186-87: ‘’These results indicated that the time of ACV incubation positively correlates with the duration of VZV inhibition’’. This correlation is not clearly supported by the findings in Figure 3. Further clarification or rephrasing may be necessary in the text to better explain this relationship

P10l194: Fig4B is not referred in the text.

P15l315: change:’’ the top and center panels in Figure 8’’ by ‘’ the top and center panels in Figure 8A’’

P15l319: change H3K3me3 by H3K4me3

Reviewer #3: 1) Figure 1 and associated text: it is unclear how many differentiation tests this is from. Statistical analysis is needed.

2) Multiple breaks in a graph are difficult to compare across samples. A log scale would be easier to follow.

3) Citations of figure 4 b-d are incorrect in the text between lines 195-198.

4) Figure 3 has a huge range of N for each time point (from 2 to 70), but this is not provided in the other figures. Again, thorough statistical analysis is needed to draw conclusions.

PLOS authors have the option to publish the peer review history of their article (what does this mean?). If published, this will include your full peer review and any attached files.

Reviewer #1: No

Reviewer #2: No

Reviewer #3: No
---

## [Editor Report · Decision Letter 1]

20 Jan 2025

Dear Dr. Viejo-Borbolla,

We are pleased to inform you that your manuscript 'Repression of varicella zoster virus gene expression during quiescent infection in the absence of detectable histone deposition' has been provisionally accepted for publication in PLOS Pathogens.

Best regards,

Donna M Neumann

Academic Editor

PLOS Pathogens

Robert Kalejta

Section Editor

PLOS Pathogens

Sumita Bhaduri-McIntosh

Editor-in-Chief

PLOS Pathogens

orcid.org/0000-0003-2946-9497

Michael Malim

Editor-in-Chief

PLOS Pathogens

orcid.org/0000-0002-7699-2064
---

## [Editor Report · Acceptance letter]

Dear Dr. Viejo-Borbolla,

We are delighted to inform you that your manuscript, "Repression of varicella zoster virus gene expression during quiescent infection in the absence of detectable histone deposition," has been formally accepted for publication in PLOS Pathogens.

Best regards,

Sumita Bhaduri-McIntosh

Editor-in-Chief

PLOS Pathogens

orcid.org/0000-0003-2946-9497

Michael Malim

Editor-in-Chief

PLOS Pathogens

orcid.org/0000-0002-7699-2064